# The genetic consequences of dog breed formation—Accumulation of deleterious genetic variation and fixation of mutations associated with myxomatous mitral valve disease in cavalier King Charles spaniels

Erik Axelsson[1]*, Ingrid Ljungvall[2º], Priyasma Bhoumik[3º¤a], Laura Bas Conn[1º], Eva Muren[1], Åsa Ohlsson[4], Lisbeth Høier Olsen[5], Karolina Engdahl[2], Ragnvi Hagman[2], Jeanette Hanson[2], Dmytro Kryvokhyzha[1], Mats Pettersson[1], Olivier Grenet[3], Jonathan Moggs[3], Alberto Del Rio-Espinola[3], Christian Epe[6¤b], Bruce Taillon[6], Nilesh Tawari[6], Shrinivas Mane[6], Troy Hawkins[6], Åke Hedhammar[2], Philippe Gruet[7], Jens Häggström[2], Kerstin Lindblad-Toh[1,8]

1 Science for Life Laboratory, Department of Medical Biochemistry and Microbiology, Uppsala University, Uppsala, Sweden, 2 Department of Clinical Sciences, Swedish University of Agricultural Sciences, Uppsala, Sweden, 3 Translational Medicine, Novartis Institutes for BioMedical Research, Basel, Switzerland, 4 Department of Animal Breeding and Genetics, Swedish University of Agricultural Sciences, Uppsala, Sweden, 5 Department of Veterinary and Animal Sciences, University of Copenhagen, Copenhagen, Denmark, 6 Elanco Animal Health, Greenfield, Indiana, United States of America, 7 Novartis Animal Health, St: Aubin, Switzerland, 8 Broad Institute of MIT and Harvard, Cambridge, Massachusetts, United States of America

º These authors contributed equally to this work.
¤a Current address: Scientific & IT Services, ETH Zürich, Basel Campus, Basel, Switzerland
¤b Current address: Boehringer Ingelheim Animal Health, Ingelheim, Germany
* erik.axelsson@imbim.uu.se

**Data Availability Statement:** Sequence data generated for this project are available at SRA

## Abstract

Selective breeding for desirable traits in strictly controlled populations has generated an extraordinary diversity in canine morphology and behaviour, but has also led to loss of genetic variation and random entrapment of disease alleles. As a consequence, specific diseases are now prevalent in certain breeds, but whether the recent breeding practice led to an overall increase in genetic load remains unclear. Here we generate whole genome sequencing (WGS) data from 20 dogs per breed from eight breeds and document a ~10% rise in the number of derived alleles per genome at evolutionarily conserved sites in the heavily bottlenecked cavalier King Charles spaniel breed (cKCs) relative to in most breeds studied here. Our finding represents the first clear indication of a relative increase in levels of deleterious genetic variation in a specific breed, arguing that recent breeding practices probably were associated with an accumulation of genetic load in dogs. We then use the WGS data to identify candidate risk alleles for the most common cause for veterinary care in cKCs–the heart disease myxomatous mitral valve disease (MMVD). We verify a potential link to MMVD for candidate variants near the heart specific *NEBL* gene in a dachshund population and show that two of the *NEBL* candidate variants have regulatory potential in heart-derived cell lines and are associated with reduced *NEBL* isoform nebulette expression in

under the following BioProject accession: PRJNA693123.

**Funding:** Funding from Elanco (previously Novartis Animal Health) to KLT covered the generation of WGS data and parts of EA's salary. Elanco (previously Novartis Animal Health) influenced the choice of dog breeds sequenced in this study, participated in data analysis and edited and reviewed the manuscript. Two grants from the Agria and SKK Research Foundation, one to IL (19969) and one to RH (P2011-0021), provided funding for sampling of dogs. EA was funded by a grant from the Swedish Research council (2016-03826) and a grant from FORMAS (2016-01312), both of which contributed to EA's salary. KLT is a Distinguished Professor at the Swedish Research Council (D0816101). PB's salary was funded by a Novartis postdoctoral fellowship (https://www.novartis.com/our-science/postdoc-program). With the exception of Elanco (see above) the funders had no role in study design, data collection and analysis, decision to publish, or preparation of the manuscript.

**Competing interests:** I have read the journal's policy and the authors of this manuscript have the following competing interests: Elanco (previously Novartis Animal Health) funded the generation of WGS data.

papillary muscle (but not in mitral valve, nor in left ventricular wall). Alleles linked to reduced nebulette expression may hence predispose cKCs and other breeds to MMVD via loss of papillary muscle integrity.

## Author summary

As a consequence of selective breeding, specific disease-causing mutations have become more frequent in certain dog breeds. Whether the breeding practice also resulted in a general increase in the overall number of disease-causing mutations per dog genome is however not clear. To address this question, we compare the amount of harmful, potentially disease-causing, mutations in dogs from eight common breeds that have experienced varying degrees of intense selective breeding. We find that individuals belonging to the breed affected by the most intense breeding—cavalier King Charles spaniel (cKCs)—carry more harmful variants than other breeds, indicating that past breeding practices may have increased the overall levels of harmful genetic variation in dogs. The most common disease in cKCs is *myxomatous mitral valve disease* (MMVD). To identify variants linked to this disease we next characterize mutations that are common in cKCs, but rare in other breeds, and then investigate if these mutations can predict MMVD in dachshunds. We find that variants that regulate the expression of the gene *NEBL* in papillary muscles may increase the risk of the disease, indicating that loss of papillary muscle integrity could contribute to the development of MMVD.

## Introduction

### Dog domestication and breed creation

Following initial domestication from wolves, semi-controlled breeding allowed humans to shape dog varieties; each of which were mentally and physically suited for widely different tasks. More recently, during the past 200–300 years, guidelines with the purpose of reinforcing particular desirable traits formalized this process by promoting strictly controlled breeding in closed populations. While this process created an extraordinary diversity in dog morphology and behaviour, bottlenecks and restricted gene flow associated with breed formation also resulted in loss of genetic variation and random enrichment of individual disease mutations [1]. As a consequence, relatively few, common, large effect variants are now likely contributing to high risk for certain diseases in particular breeds.

Whereas certain diseases have become prevalent in individual dog breeds, it is not clear if the recent breeding praxis affect the overall disease burden (or genetic load), nor what mechanisms may have contributed to any potential change. Depending on the mode of inheritance and the distribution of fitness effects for deleterious mutations, several mechanisms may theoretically have altered levels of genetic load during breed creation. First, theory predicts that slightly deleterious mutations with additive effects are removed less efficiently in small populations, potentially leading to an accelerated accumulation of harmful, disease associated, genetic variation as a consequence of founder events and restricted gene flow during breed formation [2]. Contrary to this, strongly deleterious recessive alleles may be exposed to selection in small populations due to increased homozygosity and therefore purged from heavily bottlenecked dog breeds [3,4]. It is also possible that artificial selection for desirable traits during breed creation could have increased the disease burden if genes targeted by selection also cause disease,

or if deleterious variants linked to variants underlying traits of interest hitchhiked to higher allele frequencies [5]. Finally, if many diseases are caused by moderately deleterious recessive mutations that are not purged, even if exposed in a homozygous state, breed formation may have caused a rise in mutational load simply due to increased cooccurrence of alleles, without altering the total number of deleterious alleles per genome (i.e. without affecting the efficacy of selection).

The initial phase of dog domestication, up until the start of recent breed formation, was associated with an estimated 16-fold reduction in effective population size (*Ne*) (from more than 32,000 to 1,640–1,980 individuals in the Basenji breed for instance), resulting in an *Ne* five times smaller than that of the extant wolf population [6]. Congruent with relaxed negative selection and accumulation of harmful mutations with additive effects during dog domestication, the proportion of amino acid changing to silent site variation is higher in extant dogs relative to in extant wolves [7–9] and breed dogs carry 2–3% more derived, potentially deleterious, alleles at evolutionarily conserved amino acid changing sites than wolves [5]. Dogs also carry more derived potentially deleterious alleles in a homozygous state than wolves indicating that the disease burden in dogs has increased regardless of whether disease alleles are predominantly additive or recessive [5].

Although levels of genetic load thus most likely increased during dog domestication, it is still not known how breed formation affected this process. Breed formation led to further reductions in *Ne* but the magnitude and duration of the breed bottlenecks varied substantially [10–12] (Dreger et al [12] noted that recent *Ne* for 80 analysed dog breeds span a five-fold range) arguing that different breeds may have continued to accumulate deleterious variants at different rates. In potential agreement with this, the ratio of amino acid changing to silent site heterozygosity correlate negatively (although weakly) with neutral heterozygosity across dog breeds indicating that selection may have become further relaxed in breeds with particularly small population sizes [5]. This observation may however not necessarily reflect differences in the effectiveness of selection, as it may well have been influenced by drift and mutation during breed bottlenecks [3], and so far no unbiased comparisons of the amount of potentially deleterious genetic variation (mutational load) between breeds have been made.

Here we generate whole genome resequencing data from 20 individuals from each of eight common dog breeds to investigate if levels of mutational load differ between breeds. We estimate that dogs from the most heavily bottlenecked breed investigated here—cavalier King Charles spaniel (cKCs)—carry 6–13% more derived, potentially deleterious alleles, than other dog breeds investigated here at sites displaying high sequence conservation. This represents the first clear documentation of a continued accumulation of mutational load during dog breed formation indicating that the breeding process may have been associated with an increased overall disease burden.

The most common cause for veterinary health care in cKCs is the acquired heart disease myxomatous mitral valve disease (MMVD) [13]. In the general dog population MMVD accounts for approximately 75% of all cardiac disease cases [14] and a homologous disease affect approximately 2% of the human population [15]. The disease is characterized by heart valve degeneration that primarily affect the mitral valve, which separates the left atrium from the left ventricle [16]. The valvular degeneration overtime leads to mitral valve leakage (mitral regurgitation or MR), which may lead to chronic volume overload as the disease progresses. Affected dogs can usually compensate the MR for years; but eventually the heart might become incapable of meeting the increased workload, and congestive heart failure with pulmonary congestion and oedema develops.

Although the process of the degenerative valve remodelling has been well described, the underlying causes and mechanisms involved in the development and progression of the

disease remain poorly characterized [17]. Pedigree analyses and a succesfull breeding program to decrease disease prevalence suggest that MMVD is heritable in both cKCs [18,19] and dachshund [20] in a manner suggesting polygenic inheritance. In line with this, MMVD is more prevalent and has an earlier onset in small to medium-sized dogs, such as dachshund, miniature poodles and Yorkshire terriers. It is particularly common and has an unusually early onset in cKCs, in which approximately 50% of individuals aged 6–7 years and nearly 100% of cKCs aged 11 will have developed the disease [18]. Familial transmission has also been described in humans and so far genetic variants in five human genes have been associated with increased risk of MMVD, including genes affecting cell migration and autophagy during valve development [15,21–23]. In dogs, one GWAS [24] identified two loci (CFA13 and 14) associated with early disease onset in cKCs. However, these loci are unlikely to explain all of the exceptionally high prevalence of MMVD in cKCs, arguing that relaxed negative selection, drift and/or positive selection at genes underlying both desirable traits and disease risk, may have led to a substantial rise in frequency of one or several additional risk alleles that now contribute to the high prevalence of MMVD in cKCs.

To identify alleles that potentially contribute to the high risk of MMVD in cKCs we scanned the sequence data generated here for high frequency derived variants in cKCs affecting evolutionarily conserved sites in genes with known roles in heart development and function or in processes involved in MMVD pathogenesis. We identified 10 such candidate variants, four of which affect nonsynonymous sites in four different genes (*LPHN2*, *SORBS2*, *HTR1F* and *HDGFL1)*, and six of which affect potential regulatory sites near the heart specific *NEBL* gene. To further assess whether any of these variants affect disease susceptibility in dogs, we screened a second breed that is also affected by MMVD—dachshund—for clinical and echocardiographic signs of the disease, and tested if candidate variant genotypes could be used to predict a diagnosis of MMVD. We found that three of the variants near *NEBL* were associated with disease onset at an early age. To explore the regulatory potential of these variants we then studied allele specific effects on variant-protein interactions in heart derived cell lines and compared mRNA levels of nearby genes in heart tissues from canine donors with different genotypes. We found that two of the variants near *NEBL* show evidence of allele specific protein interactions and that one of the disease-associated alleles is associated with reduced *NEBL* muscle isoform Nebulette expression in papillary muscle, but not in left ventricular wall, nor in mitral valve. Considering the pivotal role of the papillary muscle in maintaining mitral valve geometry and integrity to prevent systolic mitral valve prolapse, our results may indicate that papillary muscle weakening could constitute a significant component of the MMVD disease aetiology.

## Results

### Sequencing and variant discovery

A total of 1,708 Gb of sequence data from 160 dogs was mapped to the canine reference genome CanFam3.1 [25], corresponding to an average sequencing depth of 4.5x per individual (Table 1). Throughout the entire dataset we discovered 11,899,463 single nucleotide variants (SNVs), 2,933,658 insertion and deletions (InDels), 5,392 large deletions and 3,924 copy number variants (CNVs) (S1 and S2 Data). Our study design was aimed at discovering most variants >5% allele frequency in eight common dog breeds. In line with this goal, we estimate that we detect 97.8% of all biallelic SNPs with a minor allele frequency > = 5% in cKCs, at a false discovery rate (FDR) of 1.5% (Fig 1 in S1 Text and Table 1 in S1 Text). As expected, within breeds we observe a significant rise in the cumulative proportion of variants detected with every additional individual analysed up to the first five genomes (at which point 73–76% of all

**Table 1. Summary of sequencing and variant discovery.**

| | beagle | cKCs | German s. | golden r. | Labrador r. | s. poodle | Rottweiler | WHwt | All dogs |
|---|---|---|---|---|---|---|---|---|---|
| Samples | 20 | 20 | 20 | 20 | 20 | 20 | 20 | 20 | 160 |
| Tot. mapped (Gb) | 248 | 227 | 185 | 210 | 212 | 245 | 178 | 202 | 1708 |
| Av. mapped depth | 5,18 | 4,75 | 3,86 | 4,39 | 4,43 | 5,12 | 3,73 | 4,23 | 4,46 |
| SNVs | 7 881 176 | 5 478 591 | 7 034 171 | 6 858 417 | 7 675 446 | 7 387 846 | 6 220 187 | 6 327 639 | 11 899 463 |
| Private SNVs | 362 082 | 143 143 | 260 151 | 184 044 | 266 541 | 233 650 | 184 552 | 158 208 | |
| Fixed der. SNVs | 442 892 | 989 250 | 606 908 | 646 843 | 479 599 | 533 784 | 792 570 | 765 451 | |
| INDELs | 1 534 682 | 1 114 780 | 1 376 256 | 1 354 240 | 1 497 205 | 1 451 094 | 1 242 116 | 1 262 237 | 2 933 658 |
| Fixed der. INDELs | 68 329 | 150 177 | 94 209 | 99 232 | 73 445 | 81 462 | 120 893 | 116 492 | |
| Large deletions | 4013 | 3093 | 3412 | 3554 | 3880 | 3788 | 3224 | 3292 | 5392 |
| Private large deletions | 118 | 73 | 80 | 55 | 75 | 77 | 74 | 73 | |
| CNVs | 1096 | 828 | 1413 | 1242 | 1493 | 1014 | 1430 | 1444 | 3924 |
| Private CNVs | 97 | 62 | 128 | 75 | 112 | 72 | 134 | 134 | |

Number of samples, total mapped bases and average mapping depth per breed (cKCs–cavalier King Charles spaniel, WHwt–West Highland white terrier). *SNVs, INDELs, Large deletions and CNVs*–number of single nucleotide variants, insertion and deletions, large deletions and copy number variants, respectively, detected per breed, and in the complete data set. *Private SNVs, Private large deletions and Private CNVs*–number of SNVs, Large deletions and CNVs only found to be segregating in a specific breed. *Fixed derived (der.) SNVs and Fixed derived (der.) INDELs*–number of SNV and INDEL sites, respectively, segregating in the complete data set that were fixed for the derived allele in each breed.

variants have been detected in all breeds), followed by minor gains in power with the addition of more individuals (Fig 2 in S1 Text). Ninety percent of all variants were detected in 10 to 12 individuals, depending on which breed that was analysed.

The breeds investigated here are approximately equally distantly related (Fig 3 in S1 Text), each representing a separate European breed clade, with the exception of golden and Labrador retriever that are more closely related, reflecting their common retriever origin [26]. Congruent with expectations from the multiple bottlenecks believed to be associated with dog domestication and breed formation, we observe an excess of high frequency derived variants and a lack of rare variants throughout all breeds analysed here (Fig 4 in S1 Text) and formal site frequency-based demographic modelling suggests that most breeds have undergone at least two bottlenecks (Fig 5 in S1 Text). Levels of genetic variation however vary substantially between breeds, with relatively high levels in beagles and Labradors, moderate levels in standard poodles, German shepherds and golden retrievers, and low levels in West Highland white terriers (WHwt) and Rottweilers, indicating that the duration or intensities of bottlenecks varied significantly across breeds (Table 1). cKCs carry significantly fewer segregating sites (Table 1) and is less heterozygous (Table 2 in S1 Text) than any of the other breeds, a pattern that is consistent across all categories of variants (Table 1). In line with a more extreme bottleneck during the history of cKCs, linkage disequilibrium extends over longer distances (Fig 6 in S1 Text) and more regions are homozygous (Fig 7 in S1 Text) in cKCs relative to in other breeds.

## Mutational load

To assess whether breeding practices may have affected levels of mutational load in dogs we evaluated if the burden of deleterious genetic variation varies between breeds and, if so, whether more heavily bottlenecked breeds suffer from more load than other breeds. To this end we first assumed an additive model for allelic effects and used the $R_{A/B}$ statistic to compare the total numbers of potentially deleterious mutations in all breeds analysed here [3]. This statistic measures the number of derived mutations observed in breed A that are not observed in

breed B and vice versa, and then calculates the ratio, $R_{A/B}$, of the two numbers. Any deviation from the expected $R_{A/B} = 1$ would imply that mutations have accumulated at different rates in the two breeds since they last shared a common ancestor. To be able to assess the impact of a potential relaxation of selection on mutations with different fitness consequences we calculated the $R_{A/B}$ statistic for three different categories of sites (*synonymous*, *nonsynonymous* and *loss of function (LoF,)*) and sites evolving under different evolutionary constraint (*moderately conserved*: 2<*PhyloP* < = 5 and *highly conserved*: *PhyloP*>5) using 100 vertebrates do determine constraint. Standard errors for $R_{A/B}$ were estimated using weighted-block jackknife.

As expected for sites evolving under neutrality and with no mutation rate differences between breeds, for synonymous sites we observe no significant deviations from expectations ($R_{A/B} = 1$) for any breed pair analysed here (Table 3 in S1 Text). We also see no differences in the numbers of derived nonsynonymous mutations between these breeds (Table 4 in S1 Text). Rottweiler display a slight increase of mutations at *LoF* sites relative to both WHwt and standard poodle, but these deviations are not significant after correcting for multiple testing (Table 5 in S1 Text). For *moderately conserved* sites we see an accumulation of derived mutations in standard poodles relative to in WHwt, German shepherd and golden retriever, and this deviation withstands corrections for multiple tests in the golden retriever comparison ($R_{GR/POODLE} = 0,98$; p = 0,0002; Table 6 in S1 Text). The most conspicuous evidence for accelerated accumulation of mutational load during breed formation is seen in cKCs, which carry 6.4% to 12.5% more derived mutations at *highly conserved sites* relative to the other breeds (Fig 1A). The increase is significant in all comparisons except that with standard poodle and withstands correcting for multiple tests when cKCs is compared to Rottweiler ($R_{CKCS/ROTTWEILER} = 1,12$; p = 0,0003; Fig 1B).

Two processes may potentially have contributed to the increase in derived deleterious variation in cKCs: relaxed negative selection or hitchhiking of deleterious alleles with variants under selection. If relaxed negative selection was responsible for the observed increase in

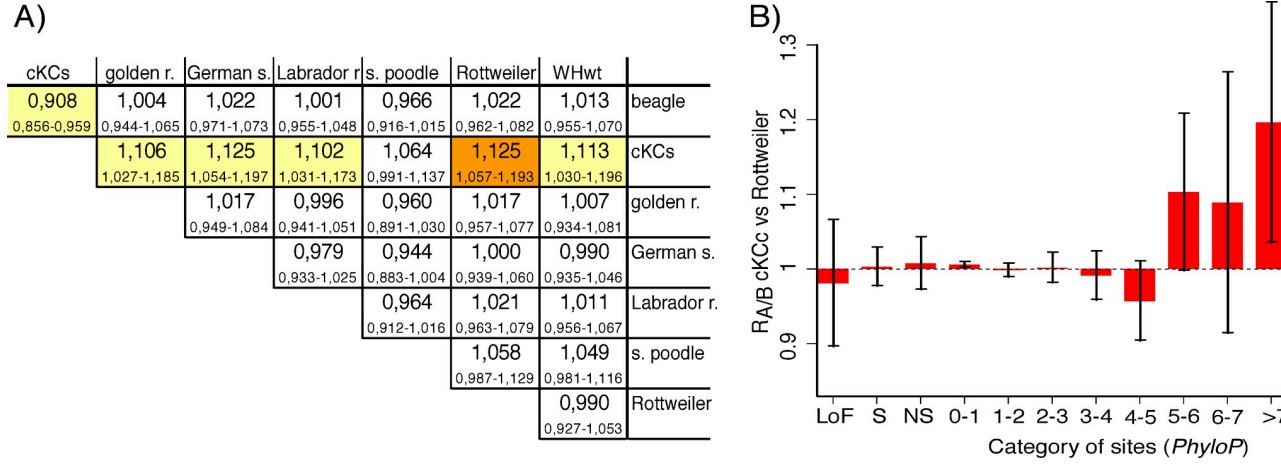

**Fig 1. Accumulation of deleterious alleles at highly conserved sites in cKCs. A)** Relative abundance of derived potentially deleterious alleles, $R_{A/B}$, at highly conserved sites (*PhyloP*$_{100\ vertebrates}$>5) for all pairwise breed comparisons. Each square presents the abundance of potentially deleterious alleles in the breed indicated at the end of the row relative to in the breed stated at the start of the column. A value below 1 indicates that there are more deleterious alleles in the breed representing the column relative to in the breed representing the row and vice versa. The range in each square represents 95% confidence interval for $R_{A/B}$. Color highlighting refers to $R_{A/B}$ values that deviate significantly from 1 (yellow), including after correcting for multiple testing (orange). **B)** Relative abundance of derived potentially deleterious alleles, $R_{A/B}$, in cKCs compared to in Rottweiler, at loss of function- (*LoF*), synonymous (*S*), nonsynonymous (*NS*) sites and in bins characterized by increasing nucleotide conservation measured using *PhyloP* (*Category of sites*). $R_{A/B}$>1 indicates that there are more deleterious alleles in cKCs than in Rottweiler. 95% confidence intervals for both panels were identified using Weighted Block Jackknife.

additive load we hypothesized that the relative increase in load in cKCs should be proportional to the difference in effective population size, as approximated based on the number of segregating sites, in each breed pair examined. However, we found no such association (Table 7 in S1 Text). To test whether the rise in load may be due to hitchhiking we masked regions of the genome that display signs of recent selection in cKCs (corresponding to 5% of the genome) and re-estimated $R_{A/B}$ at highly conserved sites in all breed pairs. Although variance increased as a consequence of the removal of data, the relative abundance of derived mutations at highly conserved sites remained almost unaffected in cKCs (Table 8 in S1 Text) indicating that hitchhiking may not have contributed substantially to the accumulation of load during cKCs breed formation.

We next compared the genetic load in breeds assuming recessive, rather than additive, effects of deleterious alleles. For this purpose, we calculated the $R^2$ statistics (which represents a modification of the above used $R_{A/B}$, [3]) to contrast the number of derived, potentially deleterious alleles found in a homozygous state in different breed pairs. We now noted significant differences in the relative amount of homozygous derived alleles across all categories of sites (including at *synonymous* sites) for almost all breed comparisons (Tables 9–13 in S1 Text) and the magnitude of the relative proportion of homozygous derived alleles in breeds correlate with differences in levels of genetic variation between breeds (Table 7 in S1 Text). There were for instance more derived alleles in homozygous state in cKCs relative to in beagle than relative to in WHwt. In cKCs, this correlation exists for all categories of sites (i.e. both sites with and without potential function), indicating that demographic effects (as selection is expected to target functional sites) are the main contributing factor to the observed differences in potential homozygous load between breeds. The relative accumulation is however greatest at *highly conserved sites (synonymous ($R^2_{cKCs/beagle} = 1,38$); nonsynonymous ($R^2_{cKCs/beagle} = 1,34$); moderately conserved sites ($R^2_{cKCs/beagle} = 1,42$); highly conserved sites ($R^2_{cKCs/beagle} = 1,63$);* (Table 7 in S1 Text)), arguing that relaxed negative selection also likely contributed to increased recessive load in cKCs.

## MMVD

Depending on the dominating mode of genotype effects for disease causing variants, accumulation of deleterious alleles due to relaxed negative selection and allele frequency shifts due to purely demographic effects may hence have contributed to a rise in the overall disease burden in cKCs relative to most other dog breeds studied here. We next hypothesized that one or several of the many high frequency-derived alleles in cKCs are likely to contribute to the exceptionally high risk of developing MMVD in this breed [13] and asked whether the sequencing data generated here could be used to identify potential MMVD risk variants. To address this question, we calculated average pairwise $F_{ST}$ for all variants across all breeds to identify high frequency derived alleles (*average pairwise $F_{ST} > = 0.7$*) at conserved, and thus potentially functional, sites (*PhyloP$> = 2$ for SNVs and InDels; and phastcon$>0$ for large deletions and CNVs*). We identified 1,158 SNVs and small InDels, 15 deletions and two CNVs meeting these criteria in cKCs (Table 15 in S1 Text). This contrasts strongly to just 54 SNVs and small InDels, one deletion and no CNVs, in beagle, again highlighting the massive allele frequency shifts associated with cKCs breed formation. Among all high frequency derived alleles identified throughout all breeds analysed here, we note several that have previously been shown to underlie phenotypic variation in dog (Table 15 in S1 Text), including *i)* the top most differentiated SNV in cKCs, which causes an amino acid change in the growth hormone receptor (GHR) that has been linked to body size variation [27]; and *ii)* the top most differentiated SNV in standard poodle, which leads to an amino acid shift in Keratin 71 (KRT71) that likely affects hair morphology (curly hair) [28], arguing that the approach taken here has the potential to identify functional genetic variation.

| Pos | Ref (ancestral) | Alt (derived) | Gene | SNPeff | PhyloP | | Reference allele frequency | | | | | | | | cKCs average pairwise FST |
|---|---|---|---|---|---|---|---|---|---|---|---|---|---|---|---|
| | | | | | 100 vert. | 46 mamm. | beagle | cKCs | German s. | golden r. | Labrador r. | s. poodle | Rottweiler | WHwt | |
| 16: 45 026 823 | C | T | SORBS2 | INTRON/NON_SYN | 10 | 2.8 | 1 | 0.18 | 1 | 1 | 1 | 1 | 1 | 1 | 0.82 |
| 6: 65 609 405 | T | C | LPHN2 | NON_SYN | 8.9 | 2.3 | 0.89 | 0.15 | 1 | 1 | 0.97 | 0.95 | 1 | 1 | 0.81 |
| 7: 41 245 057 | A | G | HDGFL1 | NON_SYN | 8.0 | 1.9 | 0.46 | 0 | 1 | 0.88 | 0.74 | 0.39 | 0.61 | 0.95 | 0.72 |
| 31: 273 549 | T | C | HTR1F | NON_SYN | 5.3 | 0.9 | 0.78 | 0.15 | 1 | 1 | 1 | 1 | 1 | 0.95 | 0.79 |

**Fig 2. Candidate MMVD risk variants affecting genes with known function of potential relevance to MMVD pathology.** Four variants that affect *highly conserved sites* (PhyloP$_{100\text{vertebrates}}$ >5) located within 5 Kb up- and downstream of genes with known function of potential relevance to MMVD pathology [17,30–32]. *Ref (ancestral)*: ancestral reference genome allele. *Alt (derived)*: derived alternative allele. *SNPeff*: snpeff annotation of site affected by mutation. *PhyloP 100 vert.* and *PhyloP 46 mamm*: conservation score for site based on comparison of 100 vertebrates and 46 mammals, respectively. *Reference allele frequency* is presented for each breed in study (cKCs–cavalier King Charles spaniel, WHwt–West Highland white terrier) and *Average pair wise F$_{ST}$* is presented for *cKCs*.

## MMVD coding candidate variants

We next used two additional filters to prioritize potential MMVD risk variants among high frequency derived variants in cKCs specifically. We first identified variants that affect *highly conserved sites* (PhyloP$_{100\text{vertebrates}}$ >5, n = 50) located within 5 kb up- and downstream of genes with known function of potential relevance to MMVD pathology. We identified four such potential risk variants, for which the cKCs allele is never observed in any other vertebrate studied (Fig 2). One of these variants affects *SORBS2* (annotated as nonsynonymous in *SORBS2* isoforms expressed mainly in brain, and intronic in isoforms expressed in heart according to dog RNA-seq data [29]), which encodes an adhesion junction/desmosome protein that is expressed in heart and mainly localized to the intercalated discs where it supports synchronized contraction of cardiac tissue [30].

A second, nonsynonymous, variant (N1019S) affects a gene, *LPHN2*, coding for a G-protein-coupled receptor (GPCR) known as latrophilin-2, which is expressed within the atrioventricular (AV) canal at the time that endothelial cells undergo an epithelial-mesenchymal transition (EMT) to form heart valves [31]. The resultant amino acid change is located at the start of the 6$^{\text{th}}$ transmembrane domain, which is involved in ligand binding and/or signal transduction in most GPCRs [33].

A third mutation causes an amino acid change (N82D) in the gene *HDGFL1* coding for Hepatoma-derived growth factor-like protein 1. The amino acid change affects the α-helix-bundle subdomain of the PWWP domain–a domain conferring histone and DNA binding in related proteins [34]. Variants in *HDGFL1* were recently shown to be significantly associated with atrial fibrillation (AF) [32] and nominally (p<10−5) associated with MMVD [35] in human populations.

Finally, a non-synonymous mutation in the gene coding for the HTR1F serotonin receptor causes an amino acid change (I349V) two residues away from the NPxxY motif, which is involved in GPCR activation [36]. This mutation has potential relevance to MMVD based on observations suggesting a possible causative link between serotonin signalling and MMVD [17].

## MMVD regulatory candidate variants

We then continued to prioritize potential MMVD-risk variants by cross-referencing all genes (n = 349) harbouring high frequency derived variants in cKCs that affect moderately or highly conserved sites (PhyloP$_{100vertebrates}$ or PhyloP$_{46mammals}$> = 2), with genes (n = 269) that were previously shown to be downregulated in mitral valves from canine MMVD cases relative to in healthy controls [37]. We identified five variants in three genes meeting these criteria (Table 16 in S1 Text); the *SORBS2* variant discussed above, a cluster of three intronic variants in the *NEBL* gene and one intronic variant in *SSH1*. *SSH1* codes for the slingshot protein phosphatase 1, which promotes cardiomyocyte myofilament maturation [38]. *NEBL* encodes two proteins; *i)* a nebulin like protein (Nebulette) that is abundantly expressed in cardiac muscle where it binds to actin and interacts with thin filaments and Z-line associated proteins, and hence may be involved in cardiac myofibril assembly [39]; and *ii)* LIM-nebulette coding for a widely expressed dynamic focal adhesion protein that has been shown to increase the rate of attachment and spreading of fibroblasts on fibronectin coated surfaces [40].

Guided by the two variant prioritization schemes described above we selected all four non-synonymous variants (*SORBS2*, *LPHN2*, *HDGFL1* and *HTR1F)* and the three non-coding *NEBL* variants for additional analyses aimed at exploring a potential association with MMVD-risk further (*NEBL* was prioritized over *SSH1* because of the multiple conserved variants identified in *NEBL*). Considering that more distantly located cis-regulatory variants, in addition to the three intronic variants identified, may also contribute to altered *NEBL* expression, we extended our search for candidate *NEBL* variants to a region spanning 1 Mb centred around the *NEBL* gene (chr2:11,650,000–12,650,000; Fig 3A) and identified three additional high frequency derived variants in cKCs that affect conserved non-coding variants (Table 17 in S1 Text). These variants were also selected for further analysis, making for a total of 10 candidate MMVD risk variants (Table 18 in S1 Text).

## Dachshund association analysis

To further explore a potential link to MMVD for the 10 selected candidate variants, we first assessed whether they may be associated with MMVD in another dog breed—dachshunds. The incidence of MMVD in the Swedish dachshund population is not as high as in cKCs (Table 18 in S1 Text), arguing that risk alleles that have become fixed in cKCs may be segregating at intermediate frequencies and therefore can be detectable to association analysis in dachshunds. For this purpose, we screened 122 Swedish dachshunds for clinical and echocardiographical signs of MMVD, at the Swedish University of Agricultural Sciences (SLU) in Uppsala, Sweden. Seventy-nine dachshunds were diagnosed with MMVD, among which disease severity was characterized as B1, B2 and C according to ACVIM guidelines [41], in 55 (n = 55), 18 (n = 18) and six (n = 6) individuals respectively. 43 dachshunds were considered healthy (A). The average age of the screened dachshunds was 10.0 years (range 4.8–15.9) and affected dogs were on average older than healthy controls dogs at the time of screening (mean age MMVD cases: 10.7 years vs. mean age healthy controls: 8.8 years; p = 3.8e$^{-5}$; β = 0.071; adj. R$^2$ = 0.125; *linear regression*) (Table 19 in S1 Text). We note that age predicted *graded disease status* better than simple binary *disease status* (healthy = 0 or affected = 1) (p = 6.4e$^{-6}$; β = 0.133; adj. R$^2$ = 0.15; *linear regression*) indicating that our diagnosis and disease grading method is reliable and sensitive (Table 19 in S1 Text). We found no association between sex and disease status in our dachshund sample (β = 0.03, n.s.; Table 19 in S1 Text), but note that earlier studies show clearly that MMVD tend to develop at a younger age in male compared to female dachshunds [20].

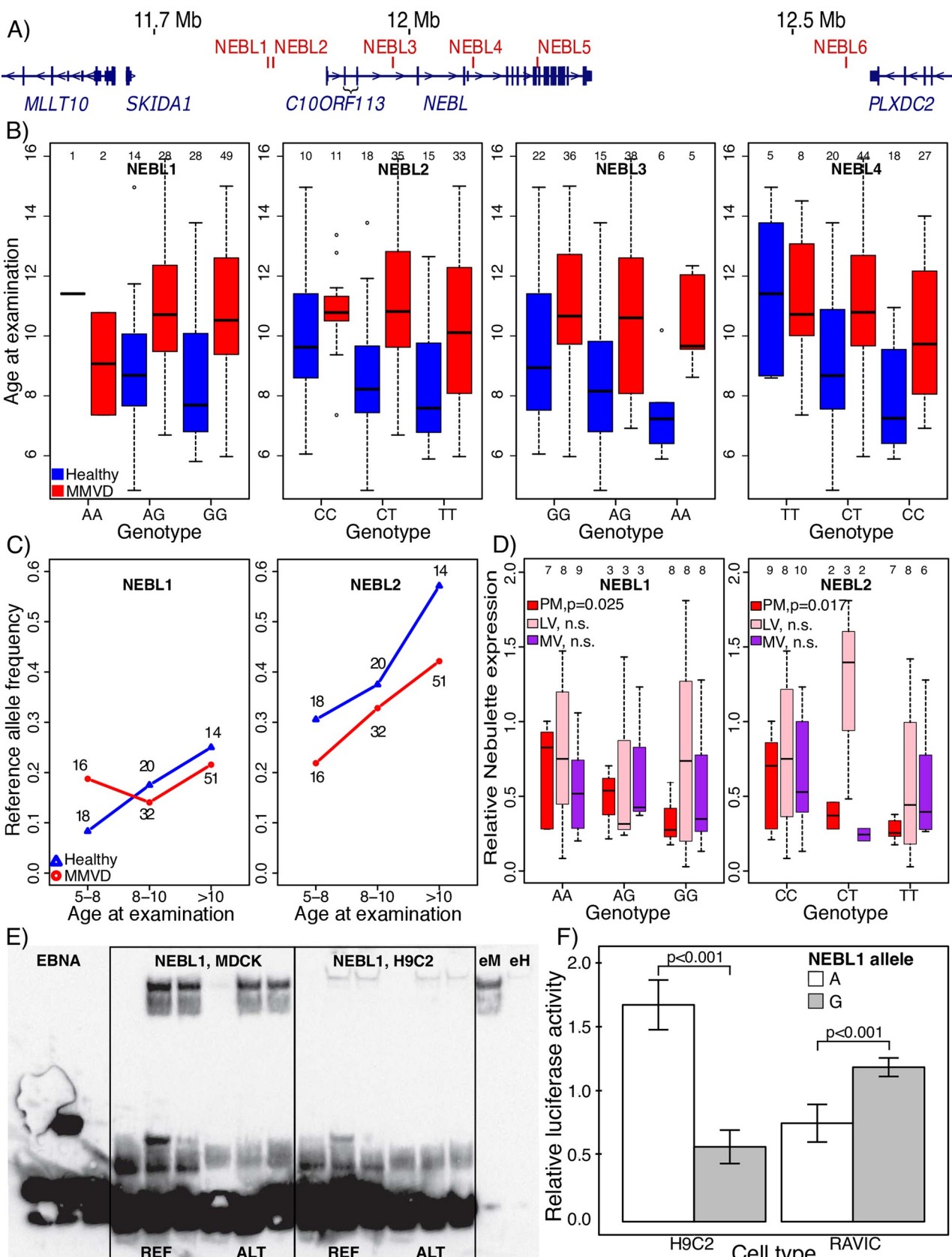

**Fig 3. Regulatory variants near *NEBL* are associated with early MMVD symptoms. A)** Chromosome 2 region spanning ~11.5 to 12.7 mega bases (Mb) harbouring *NEBL* and the six non-coding candidate variants *NEBL* 1, 2, 3, 4, 5 and 6 (red bars indicate location of variant). Gene models in blue indicate position of four genes. Tall and short bars represent coding exons and UTRs, respectively. **B)** Box plot showing age at examination and candidate variant genotype in healthy (*Healthy*, blue) and affected (*MMVD*, red) Swedish dachshunds. Sample sizes above each box. **C)** Reference allele frequencies at *NEBL* 1 and 2 in healthy (*Healthy*, blue triangle) or affected (*MMVD*, red circle) dogs aged 5–8, 8–10 or >10 years, respectively. **D)** Relative nebulette expression in papillary muscle (*PM*, red), left ventricular wall (*LV*, pink) and mitral valve (*MV*, purple) plotted for *NEBL* 1 and 2 genotypes, respectively. Sample size is indicated above each box. P-values of linear regression used to predict expression based on candidate variant genotype are shown next to the figure legends. **E)** EMSA interrogating *NEBL* 1 binding to protein extracted from rat cardiomyocytes (H9C2 cells) or canine kidney cells (MDCK cells). Binding to reference (REF) and alternative (ALT) alleles in each cell line is assayed using the following setup: lane 1 –labeled probe, no protein extract; lane 2 –labeled probe + protein extract; lane 3 –labeled probe + protein extract + excess of unlabeled probe. The three leftmost lanes depict EBNA controls. Lanes labelled *eM* (MDCK) and *eH* (H9C2) are cell extract controls. Visible shifts indicate that *NEBL* 1 reference (REF), but not alternative (ALT) allele, binds protein extracted from both cell lines. **F)** Relative luciferase activity in rat cardiomyocytes (*H9C2* cells) and rat heart valve interstitial cells (*RAVIC* cells) transfected with *luc2* containing pGL4.26 vectors and the *NEBL* 1 ancestral (A, white), or derived (G, grey), allele respectively.

We then genotyped all candidate variants in the dachshunds and noted that all, except the *SORBS2*-variant (which is fixed for the ancestral allele), segregate in the dachshund case-control population (Table 19 in S1 Text, Fig 8 in S1 Text and S3 Data) (the *SORBS2*-variant was therefore excluded from the subsequent association analysis). We assumed additive effects and used linear regression (lr) to first test whether genotype alone can be used to predict either *disease status* or *graded disease status*, but found no clear association for any of the candidate variants (Table 19 in S1 Text) (with the possible exception of *LPHN2* ($p<0.05$; $\beta$ = -0.38; adj. $R^2$ = 0.027; *lr(disease status ~ LPHN2);* however, *LPHN2* was not significant when modelling *graded disease status*). Considering that MMVD is a late-onset disease (50% of dachshunds have developed a left apical systolic murmur at 9.4 years age [20]), some individuals screened here may not yet have begun expressing clinical and/or echocardiographical signs of disease despite being risk allele carriers, arguing that the power to test for simple genotype-disease associations may be low with our data. As a first means to avoid this potential bias, we next modelled disease status as a function of *genotype*, *age at screening* and the interaction between the two factors. This analysis indicates that the genotype for *NEBL* variant 3 ($\beta$ = -0.944, $p<0.05$) and the interaction between *NEBL* 3 *genotype* and *age* ($\beta$ = 0.10, $p<0.05$; overall *NEBL* 3 model fit, $R^2$ = 0.159; Table 20 in S1 Text), as well as the interaction between *NEBL* variant 4 *genotype* and *age* ($\beta$ = 0.096, $p<0.05$; overall *NEBL* 4 model fit, $R^2$ = 0.179), are significant predictors of *graded disease status* (Fig 3B). We repeated these analyses controlling for sex, and again note that the interaction between *genotype* and *age* predicts *graded disease status* at *NEBL* 3 ($\beta$ = 0.147, $p<0.05$) and *NEBL* 4 ($\beta$ = 0.195, $p<0.01$), and now also at *NEBL* 2 ($\beta$ = 0.147, $p<0.05$) (Fig 3B and Table 20 in S1 Text). The overall model fit increased by including sex as a covariate for *NEBL* 2 ($R^2$ = 0.157) and *NEBL* 4 ($R^2$ = 0.191), but not for *NEBL* 3 ($R^2$ = 0.142).

As a further means to increase the discovery power for a late-onset, polygenic disease, such as MMVD, we next investigated whether *genotype* alone can predict simple *disease status* in samples including only young cases and old controls [42]. For this purpose, we genotyped all candidate variants in 26 additional dachshunds belonging to a retrospectively sampled Danish cohort [43,44]. Sixteen of these dogs were diagnosed with MMVD before age 10 (n = 16, avg. age = 8.2 years) and 10 dogs were healthy and at least 8 years old (n = 10, avg. age = 9.7 years). We then merged Swedish and Danish dachshund data and created four partly overlapping subsamples, each consisting of increasingly old controls and increasingly young MMVD cases. We found that *NEBL* 2 was significantly associated with MMVD *disease status* in all four subsamples (*i*: controls>8 years at screening vs. cases<10, $p<0.05$; *ii*: controls>9 years vs. cases<9, $p<0.05$; *iii*: controls>10 years vs. cases<8, $p<0.05$; *iv*: controls>11 years vs. cases<7, $p<0.01$; Table 21 in S1 Text) and note that the difference in mean *NEBL* 2 allele frequency between cases and controls widened as we compared increasingly young cases with

increasingly old controls (Fig 3C and Table 21 in S1 Text). Similar results were obtained when comparing old controls and young cases in the Swedish dachshund population only (Table 22 in S1 Text).

The disease-associated *NEBL* 2 (chr2:11,823,576), *NEBL* 3 (chr2:11,979,724) and *NEBL* 4 (chr2:12,082,890) variants reside in a region spanning approximately 200 kb. Together with nearby candidate variant *NEBL* 1 (chr2:11,816,535), located 7 kb upstream of *NEBL* 2, these variants form a block of moderate linkage disequilibrium (LD) (avg. D' = 0.89 and avg. $r^2$ = 0.29) (Fig 3A and Fig 9 in S1 Text). One or several of the alternative alleles at the *NEBL* 2, 3 and 4 variants, and potentially also at the linked *NEBL* 1 site, may hence contribute to an increased risk of developing MMVD at a younger age in dachshunds.

## Gene expression altered for different alleles

*NEBL* expression was previously found to be downregulated in a small sample of canine MMVD cases relative to in healthy controls [37]. To assess whether alleles at any of the *NEBL* variants studied here may be linked to altered *NEBL* expression, we next used qPCR to evaluate mRNA levels of *NEBL* isoform nebulette, *NEBL* isoform LIM-nebulette and the neighbouring *MLLT10* gene in tissue samples from mitral valve leaflets (n = 23), left ventricular papillary muscle (n = 23) and left ventricular wall (n = 21), from a total of 23 dogs representing eight different breeds (Table 23 in S1 Text). Expression of neighbouring genes *SKID1A* and *C10ORF113* was initially also assessed but found to be negligible in healthy heart tissue and therefore not analysed further. We found that the disease associated alternative *NEBL* 2 allele ("T") ($p < 0.05$, linear regression) and the alternative allele ("G") at the neighbouring *NEBL* 1 variant ($p < 0.05$, linear regression) are significantly associated with reduced *NEBL* isoform Nebulette expression in papillary muscle, but not in left ventricular wall, nor in mitral valve leaflets (Fig 3D). The associations between *NEBL* 1 and 2 variants and nebulette expression are consistent across replicates using two different primer pairs and in analyses using subsamples of the data defined by increasingly stringent quality thresholds (based on RNA-quality, standard deviation of technical replicates and by excluding dogs affected by dilated cardiomyopathy) (Table 24 in S1 Text). We also observed a significant association between the *NEBL* 6 variant and *MLLT10* expression in left ventricular wall that replicates in nearly all subsamples of the data (Table 24 in S1 Text).

## Electrophoretic mobility shift assays point to differential binding

The concomitant disease and gene expression association for *NEBL* variants, indicate that disrupted protein binding at regulatory sites could represent a plausible mechanism linking the associated alleles to MMVD. To explore this hypothesis we first used *in-silico* binding assays in *TRAP* [45] to search for signs of transcription factor (TF) binding for any of the linked *NEBL* 1-, 2-, 3- and 4-variants. *TRAP* indicated potential binding of several TFs to the reference alleles of *NEBL* 1 and 4 (Table 25 in S1 Text), respectively. In line with this, ENCODE data on *SCREEN* (https://screen.wenglab.org) predict that the homologous positions of *NEBL 1* and *4* in the human genome are located in cis-regulatory elements (Table 25 in S1 Text) [46]. We next used electrophoretic mobility shift assays (EMSA) to investigate potential protein binding to *NEBL* 1-, 2-, 3- and 4 *in-vitro*, using nuclear cell extracts from three different cell lines: *i)* RAVIC (a model for heart valve interstitial cell biology [47]); *ii)* H9C2 (differentiated into cardiomyocytes); and *iii)* MDCK (derived from dog kidney cells and used to finetune the protocol using probe and protein that both originate from dog). For *NEBL* 3, we documented no clear binding for either allele in any of the cell lines tested (Fig 10 in S1 Text). *NEBL* 4 bound to protein extracted from all cell lines irrespective of allele tested (Fig 10 in S1 Text). For *NEBL* 2,

binding was only unambiguously documented in valvular cell extracts, in which binding was clear for the reference allele, but only weak for the alternative allele (Fig 10 in S1 Text). *NEBL* 1 displayed no binding in valve cells (RAVIC), but the reference allele bound in both cardiomyocytes (H9C2) and MDCK cells, whereas binding for the alternative allele was abolished in cardiomyocytes and, at least partly, in MDCK cells (no binding in two replicates and only weak binding in two replicates) (Fig 3E and Fig 10 in S1 Text). *NEBL* 1 is flanked by, and in perfect LD with, another variant (*NEBL* 1.2 at chr2:11,816,545) affecting a non-conserved site located just 10 bp downstream of *NEBL* 1 (chr2:11,816,535). To assess whether the effects on protein binding observed for *NEBL* 1 are attributable to *NEBL* 1 alone, or a combination of *NEBL* 1 and 1.2, we designed four probes representing all four possible allele combinations at these sites. In line with a single effect mediated by *NEBL* 1 we only documented abolished binding for probes containing the alternative *NEBL* 1 allele using cell extracts from MDCK cells (Fig 10 in S1 Text).

## Luciferase assays

The derived *NEBL* 1 allele thus appears associated with altered Nebulette expression in papillary muscle and abolishes protein binding in cardiomyocytes, indicating that this variant may be important for gene regulation. To test this hypothesis directly we next designed a luciferase reporter assay to compare the ability of *NEBL* 1 reference and alternative alleles to drive gene expression in the two heart cell lines (RAVIC and H9C2) described above. In support of a potent gene regulatory effect, and congruent with the observed down regulation of Nebulette in papillary muscle from individuals carrying alternative *NEBL* 1 alleles, we noted significantly reduced luciferase activity (H9C2, mean relative luciferase activity *NEBL* 1 "A": 1.69, mean relative luciferase activity *NEBL* 1 "G": 0.58, p<0.001) in cardiomyocytes transfected with constructs carrying the *NEBL* 1 alternative allele (Fig 3F). Interestingly, an opposite effect on luciferase activity was noted in the valvular cell line (RAVIC, mean relative luciferase activity *NEBL* 1 "A": 0.77, mean relative luciferase activity *NEBL* 1 "G": 1.21, p<0.001; Fig 3F).

To conclude, three (*NEBL* 2, 3 and 4) out of a group of four partially linked variants (*NEBL* 1, 2, 3 and 4) near *NEBL* show associations to MMVD in dachshunds. Derived alleles at two of the linked variants, *NEBL* 1 and 2, are associated with downregulated nebulette expression in papillary muscle, and seem to abolish protein binding in extracts from cardiomyocytes and valvular cells, respectively. Congruent with these observations, replacing ancestral with derived allele at *NEBL* 1 causes a significant reduction in gene expression in cardiomyocytes when quantified using luciferase assays. One, or several, of the four linked *NEBL* variants (*NEBL* 1, 2, 3 and 4) may hence predispose dachshunds to an early onset of MMVD via a potential down-regulation of nebulette expression in papillary muscle.

## Beagle case control association test

To begin exploring whether these *NEBL* variants may affect the risk of developing MMVD in other breeds than the dachshund, we first genotyped and screened 22 beagles for clinical and echocardiographical signs of MMVD. The beagles constituted all individuals, aged >3 years (average age: 8.8 years; range 3–14.1 years), of a small population used for mainly educational purposes at SLU. Ten out of the 22 beagles were diagnosed with MMVD and affected dogs were older than healthy controls (mean age controls = 7.1; mean age affected = 10.8; p<0.01 (linear regression)). With close to 50% affected individuals, MMVD is more common in this small population than in the general Swedish beagle population in which disease frequency is moderate (Table 18 in S1 Text). While we note that allele frequencies for most *NEBL* variants were similar in the SLU population and the general Swedish beagle population, alternative *NEBL* 1 "G" was more frequent in the disease prone SLU population (SLU *NEBL* 1 "G" allele

freq. (AF) = 0.23) compared to the general Swedish population (Swedish *NEBL* 1 "G" AF = 0.09) (Table 26 in S1 Text and S3 Data). Furthermore, derived *NEBL* 1 "G" was significantly more frequent in MMVD cases (NEBL 1 $_{MMVD}$ "G" AF = 0.40) than in controls (*NEBL* 1 $_{CONTR}$ "G" AF = 0.08, p = 0.042 (linear regression)), however this association disappears when controlling for age and/or sex. Similar to *NEBL* 1, derived alleles at *NEBL* 2 and 3 tended to be more frequent in cases relative to in controls (*NEBL* 2 $_{MMVD}$ "T" AF = 0.25, *NEBL* 2 $_{CONTR}$ "T" AF = 0.08, p = 0.10; *NEBL* 3 $_{MMVD}$ "T" AF = 0.11, *NEBL* 2 $_{CONTR}$ "T" AF = 0.0, p = 0.11 (linear regression)).

### Across-breed association test

We then also genotyped the ten candidate variants (including the four nonsynonymous variants) in five additional Swedish dog populations (cocker spaniel, Norfolk terrier, Norwich terrier, Grand Danois and Dobermann pinscher) to investigate if breed-specific, average allele frequencies, at any of the candidate variants can predict breed-specific incidence of MMVD in a total set of 14 breeds (the five breeds mentioned above, the eight whole genome sequenced breeds and dachshunds). We note that allele frequencies at seven of the ten candidate variants are significant predictors of breed specific MMVD incidence (*NEBL* 1, β = -215.7, p<0.05, adj. $R^2$ = 0.34; *NEBL* 2, β = -289.3, p<0.01, adj. $R^2$ = 0.45; *NEBL* 5, β = 197.8, p<0.05, adj. $R^2$ = 0.22; *NEBL* 6, β = -326.8; p<0.001; adj. $R^2$ = 0.7; *SORBS2*, β = -398.3, p<0.001, adj. $R^2$ = 0.63; *LPHN2*, β = -342,8, p<0.001, adj. $R^2$ = 0.56 and *HTR1F*, β = -335,1, p<0.001, adj. $R^2$ = 0.57 (linear regression)) (Table 18 in S1 Text), however none of these associations remain significant after removal of cKCs from the data.

## Discussion

### Increase in additive mutational load in cKCs

We found more potentially deleterious alleles in cKCs relative to the other breeds analysed here. Considering that genome-wide patterns of genetic variation indicate that cKCs have the smallest *Ne* of the eight breeds analysed, our finding is consistent with a further relaxation of negative selection during cKCs breed creation. The effect is only observed for the most evolutionarily conserved (*PhyloP$_{100\ vertebrates}$*>5) class of sites, arguing that although cKCs have accumulated deleterious alleles that are likely to have large phenotypic effects, the total number of added mutations may be relatively low and it may hence be premature to use our results as an argument to discourage further breeding of cKCs. Nevertheless, the accumulation of deleterious variants in cKCs is sufficiently large to be detectable, which in itself is remarkable given the relatively short period of time breed bottlenecks have lasted. A similar increase in mutational load is not observed in any modern human populations, including the Greenland Inuit population which experienced a narrow bottleneck approximately 15,000 years ago [4]. The accumulation of derived potentially deleterious alleles in cKCs reflects the consequences of processes that have been ongoing during the last 200–300 years of strictly controlled breeding, but that may have started earlier given that most breeds compared here belong to separate breed groups that likely started to diverge several hundred years earlier (see discussion below on the origin of the cKCs breed). Hitchhiking does not seem to have contributed significantly to the rise in mutational load in cKCs as the relative abundance of derived deleterious alleles at highly conserved sites in cKCs remain practically unchanged after removal of regions under potential selection. Likewise, the relative increase in deleterious variants in cKCs is almost constant (10–13% more derived alleles at the most conserved class of sites) irrespective of the breed used for comparison, i.e. there is no apparent correlation between relative amounts of mutational load and differences in overall levels of genetic variation between breeds, as may have been expected if load

accumulated as a consequence of less efficient selection in small populations. We have no simple explanation to this observation but speculate that the intensity and duration of the bottleneck(s) leading to breed formation may have been extreme in cKCs relative to in most other breeds. This hypothesis is consistent with previous array-based analyses in which cKCs rank among the ~10 least variable breeds out of 80 analysed [11,26] and combinations of both long-range linkage disequilibrium and extensive runs of homozygosity in cKCs indicate that both older population contractions and recent inbreeding may indeed have been pronounced (Figs 6 and 7 in S1 Text) [11]. Likewise, site frequency-based demographic inferences suggest that the more recent of the two inferred bottlenecks associated with dog domestication may have been particularly narrow in cKCs (Fig 5 in S1 Text). Historical records suggest that small spaniel-type dogs have existed for at least 1,000 years and hint at a history involving serial bottlenecks. Small spaniels were popular at royal courts throughout Europe and Asia for several hundred years [48] and the appearance of cKCs in particular is very similar to toy spaniels kept by the royal Stuart family (rulers of Scotland, England and Ireland), including King Charles II (1630–1685). Over time a brachycephalic variety of these spaniels became the dominating type, but eventually the old, more long-nosed type was restored by a small breeding initiative starting in the 1920's. This rescued variety however suffered a serious decline in numbers during the second world war before becoming recognized as a breed of its own (cKCs) in 1945.

Our results also indicate a relative accumulation of deleterious genetic variation in standard poodles. Similar to in cKCs, standard poodles tend to have more deleterious alleles in the most conserved class of sites, but it is in the larger class of moderately conserved sites that we note a significant, 1–2%, increase in deleterious alleles relative to several of the other breeds. Interestingly, this holds true also for the comparison with WHwt, which we estimate should have a smaller effective population size than standard poodles, again indicating that our observations may not fit simple expectations based on a relaxed negative selection in small populations. Furthermore, we observe no differences in potentially deleterious alleles at *LoF* sites between breeds, arguing that recessive effects may not be important, or alternatively, that we do not define this category of sites with precision. Nevertheless, to the extent that disease alleles indeed are recessive, increased derived homozygosity across all categories of sites (synonymous, nonsynonymous, *LoF* and conserved) in heavily bottlenecked breeds, indicate that purely demographic effects may potentially have contributed to an increase in mutational load, proportional to the population size contraction during breed creation.

To conclude, independent of the dominating effect of disease mutations, we demonstrate the first unequivocal example of an increase in mutational load as a consequence of dog breed formation. Although this effect is most evident in one of the most heavily bottlenecked breeds (cKCs) known, we find no simple relationship between overall levels of genetic variation and additive load among the eight breeds analysed here. Complex interplays between different modes of selection (dominant, additive, recessive), selection coefficients, hitchhiking, and extent and duration of bottleneck(s) may potentially modulate mutational load such that no simple relationships can be expected. Quantification of load in more breeds with distinctly different demographic histories and at precisely defined categories of sites (such as conserved sites in monogenic disease genes) will be valuable to deepen our understanding of the consequences of selective dog breeding on accumulation of load.

## Both coding and regulatory mutations may predispose to development of MMVD at a young age

We hypothesized that one or several of the derived potentially deleterious alleles in cKCs may predispose cKCs to an early onset of MMVD, and used proximity to genes with known heart

function to single out 10 particularly interesting candidate variants, which we tested for potential associations with MMVD in a Swedish dachshund population. Four of the candidate variants affect highly conserved nonsynonymous sites in four separate genes (*SORBS2*, *HDGFL1*, *HTR1F* and *LPHN2*) and the remaining six are conserved noncoding variants near *NEBL*. Among the genes with coding mutations we considered *LPHN2* of potential interest for MMVD because it is expressed within the atrioventricular (AV) canal at the time endothelial cells undergo EMT to form heart valves. Knock-down of *LPHN2* in AV canal cultures inhibits lateral cell migration (cells fail to detach from each other) and transformation of cells into a fibroblastic shape and results in formation of unusually small numbers of mesenchymal cells during EMT [31]. *HDGFL1* could be relevant to canine MMVD given that it was recently found to be nominally associated with human MMVD [35]. *SORBS2* encodes an adhesion junction/desmosome protein localized at actin stress fibres and intercalated discs where it supports synchronized contraction of cardiac tissue. Loss of the SORBS2 protein in mouse results in phenotypes characteristic of human arrhythmogenic right ventricular cardiomyopathy (ARVC), including dilated right ventricle (RV), RV dysfunction, spontaneous ventricular tachycardia (VT), and premature death [30]. There is however so far little evidence linking *SORBS2* to heart valve function, but we nevertheless consider the *SORBS2* mutation interesting given that it affects the top most conserved site of all highly differentiated cKCs variants and because *SORBS2* expression is reduced in mitral valves from cKCs affected by MMVD. Serotonin receptor HTR1F may be relevant given evidence that altered serotonin signalling could be causatively linked to MMVD. Findings supporting this link include *i)* increased circulating serotonin concentrations in dogs affected by MMVD [49], *ii)* higher serotonin concentrations in mitral valves from dogs with MMVD relative to in healthy dogs, *iii)* development of valve injuries secondary to serotonin producing tumours and serotonergic drugs in human and *iv)* triggering of valve lesions in animal models exposed to exogenous serotonin administration [17]. So far, most evidence indicate that activation of 5HT2B (encoded by *HTR2B*) may be underlying the observed effects of serotonin signalling on valve disease [17] and although *HTR1F* is abundantly expressed in several heart tissues (including atrium, ventricle wall, epicardium and coronary artery) [50] it does not seem to be expressed in interstitial cells isolated from mitral valve [51]. None of the four nonsynonymous variants described above were associated with MMVD in dachshund. However, we note that these variants either do not segregate (*SORBS2*), or segregate at low allele frequencies in dachshunds (*LPHN2* MAF = 0.03; *HTR1F* MAF = 0.03; *HDGFL1* MAF = 0.13), arguing that the lack of associations could be due to low detection power. It is thus premature to rule out potential contributions to MMVD risk from the non-synonymous candidate variants based on our analyses and we suggest these variants should be investigated in MMVD case-controls cohorts sampled from other breeds as they may contribute to risk in all cKCs dogs.

In contrast, all six *NEBL* variants segregate at intermediate frequencies (MAF = >0.2) and genotypes at three of these (*NEBL* 2, 3 and 4) predicted the occurrence of MMVD signs using models controlling for either *age at examination*, or both *age at examination* and *sex*. Derived alleles at one or several noncoding variants near *NEBL* may hence predispose carriers to develop MMVD at a relatively early age. Consistent with this, univariate analysis showed that *NEBL* 2 genotype predicted MMVD in subsamples of the dachshund population that only included increasingly young affected individuals and increasingly old healthy controls, but not in the complete dachshund population (age range: 5–16 years). Comparisons of young cases and old controls will increase discovery power for late onset polygenic diseases, such as MMVD, [42] as it removes biases introduced by *i)* risk allele carriers that are too young to have started expressing signs of disease, *ii)* old cases in which MMVD partly may have been triggered by mechanical wear rather than high polygenic risk and *iii)* a paucity of old risk allele

carriers due to premature deaths associated with the disease [52]. Congruent with a potential link between regulatory *NEBL* variants and MMVD we also note that the derived *NEBL* 1 allele predicted echocardiographic signs of MMVD using univariate analysis in a small Swedish beagle population (n = 22). However, this association did not replicate when comparing old cases and young controls, nor when controlling for *sex* or *age at examination*. Finally, although our across-breed association analysis also indicated a potential link to MMVD for several of the *NEBL* variants studied here, we stress that this signal was mostly driven by cKCs and that this analysis should be replicated using a larger sample of breeds to reliably test for a potential major association to MMVD throughout the general dog population.

 *NEBL* encodes two proteins, both of which may potentially affect normal heart function. The first protein, nebulette, is specifically expressed in cardiac muscle, where it, similar to nebulin in skeletal muscle, binds and stabilizes a core region of thin filament actin and anchors the thin filament to the Z-line by binding to several Z-line associated proteins, including desmin [39,53]. By connecting the thin filaments to the Z-disc nebulette is hence likely important for thin filament and Z-disc stability and myofibrillar connectivity [54]. The second protein, LIM-nebulette (or LASP-2), is expressed widely, including in neuronal tissue. It is found in actin rich structures such as focal adhesions where it interacts with zyxin [55]. LIM-nebulette has been shown to regulate dendritic spine development and synapse formation and is thus likely to be important for neuronal circuitry formation [56]. It has also been demonstrated to affect fibroblast cell migration [57]—a process that has previously been demonstrated to be of relevance to proper heart valve development [15].

 The *NEBL* variants studied here qualified as candidate variants due to a previously documented downregulation of *NEBL* expression in dogs affected by MMVD [37]. By demonstrating that derived alleles at two of the disease associated candidate variants (*NEBL* 1 and 2) were associated with reduced *NEBL* expression in canine heart, our analyses indicate that the previously documented *NEBL* downregulation could potentially be cause, rather than effect, of MMVD, and that altered gene regulation could hence represent a plausible molecular mechanism underlying the disease associations documented here. By furthermore demonstrating that the disease associated allele was specifically linked to reduced nebulette-, rather than LIM-nebulette expression, in papillary muscle, our analyses also indicate that altered nebulette (rather than LIM-nebulette) function may be relevant to MMVD pathology and that the nebulette dysfunction may primarily affect papillary muscles (rather than the left ventricular heart wall or the mitral valve itself).

## Investigating the regulatory potential of the disease associated NEBL variants

To evaluate the regulatory potential of the disease associated variants we used EMSA to assess protein binding in cell extracts from rat cardiomyocytes and valve cells, respectively. We included *NEBL* 1 in this analysis, although it was not directly associated with increased disease risk in dachshund since it *i)* may be associated with MMVD in a small beagle population, *ii)* is associated with altered nebulette expression in dog, *iii*) displays higher evolutionary constraint than the other *NEBL* variants and *iv)* forms a block of moderate LD with the three disease associated, *NEBL* variants (*NEBL* 2, 3 and 4) in dachshund, indicating that *NEBL* 1 may potentially represent a causative variant itself. If so, the lack of a direct disease association for *NEBL* 1 in dachshund could potentially be explained by an unbalanced genotype representation, with a near complete lack of individuals carrying the ancestral allele in a homozygous state. In line with the results of the expression analyses, the EMSAs indicated no difference in protein binding for *NEBL* 3 and 4 alleles, but documented evidence of disrupted protein binding for

derived, relative to ancestral alleles, at both *NEBL* 1 and 2. Furthermore, whereas *NEBL* 2 only bound to valvular cell extracts, binding to *NEBL* l was observed specifically in cardiomyocytes, indicating that the association between papillary nebulette expression and *NEBL* 1 and 2 may ultimately be driven by altered protein binding to *NEBL* 1. In direct support of a tissue specific regulatory potential for NEBL 1, we found that the ancestral allele at *NEBL* 1 produced a two-fold rise, relative to the derived allele, in luciferase activity in cardiomyocytes, but not in valve cells.

## Limitations of our analyses and potential consequences of nebulette dysfunction

A lack of correction for multiple testing limits the strength of each individual association test performed here, however we argue that the result of each test combined, including *i*) across breed allele frequency differences, *ii*) within breed MMVD associations in both dachshunds and beagle, *iii*) associations to nebulette expression and *iv*) evidence for regulatory potential, indicate that one or several of the NEBL mutations studied here may predispose carriers to develop MMVD at an early age. Several coding mutations in nebulette have previously been identified (and found to be causative) in human patients with dilated cardiomyopathy (DCM) and endocardial fibroelastosis [54]. Loss of endogenous nebulette expression in chicken cardiomyocytes resulted in a destabilization of thin filaments, or the entire myofibril, and a drastic reduction in cardiomyocyte beating frequency [58]. Knock-out of *Nebl* in mice led to widened Z-lines and increased expression of stress responsive genes in heart muscle [59]. However, unlike in chicken cardiomyocytes, loss of nebulette did not seem to alter cardiac function in mouse (at least not at the age of 9 months). Nevertheless, the combined observations of Z-line widening and upregulation of stress factors in *Nebl*$^{-/-}$ mice, together with the more severe loss of function phenotype in chicken cardiomyocytes lacking nebulette, argue that nebulette may play a similar role in maintaining cardiomyocyte Z-line integrity to that observed for nebulin in skeletal muscle, in which loss of nebulin render muscles susceptible to eccentric contraction-induced injury [60], thus suggesting that nebulette potentially may serve to protect heart muscle from injury during physical load.

## The role of papillary muscles for mitral valve function

The papillary muscles are part of the mitral valve apparatus and attach the two mitral valve leaflets to the left ventricular heart wall via the cordae tendineae (CT) [61]. During contraction of the left ventricle the papillary muscles ensure that enough force is applied to the mitral valve to hinder the valve from prolapsing into the left atrium thereby avoiding MR [62]. Although it is clear that acute papillary muscle rupture following on myocardial infarction or trauma will cause severe MR [63], it is less well understood if mild or moderate papillary muscle dysfunction by itself may lead to mitral valve prolapse and MR. For instance, in humans with prior myocardial infarction MR is more common in patients suffering from combined anterior and posterior, relative to normal or isolated, papillary muscle dysfunction [64], but there is little evidence that isolated ischemic papillary muscle dysfunction is associated with MR [65]. Furthermore, whereas papillary muscle shortening during systole is reduced in humans suffering from mitral valve prolapse [66], as well as in human DCM patients with MR, relative to in DCM patients without MR [67], it is not clear whether altered papillary muscle movement is the cause or effect of these findings [68]. In dogs intramyocardial arteriosclerosis and fibrosis is more abundant in individuals affected by MMVD than in healthy controls [69,70]. This fibrosis ranges from mild interstitial changes to bigger areas of confluent replacement type fibrosis, but the changes are more pronounced in the papillary muscle relative to in other parts

of the heart [69,70]. Interestingly, the extent of the papillary muscle fibrosis predicts mitral valve function (as quantified based on regurgitant blood flow) in dogs with MMVD, indicating that papillary muscle fibrosis may contribute to MR in dogs with MMVD [69]. In light of these observations and the previously documented causative role of amino acid changes in nebulette for endocardial fibroelastosis in humans [54], it is tempting to speculate that our finding that regulatory *NEBL* alleles may be associated with MMVD via their effect on nebulette expression in papillary muscle could underlie the observed link between papillary muscle fibrosis and deteriorating mitral valve function in dogs affected by MMVD. If so, could loss of papillary muscle integrity also contribute to myxomatous remodelling of mitral valve leaflets? Given that a sufficiently weakened papillary muscle may hypothetically be unable to apply enough traction to balance normal pull from the mitral valve during systole and maintain the dome-shaped geometry of the valve, it is plausible to assume that altered mitral valve shape, leading to a flattening of the valve [71], and later leaflet prolapse [72] and MR may follow. A shift from normal leaflet curvature and a saddle-like shape of the mitral annulus is known to lead to increased peak leaflet stress [73], which in turn triggers signalling pathways that may contribute to MMVD [68,74] and deposition of superimposed tissue on the valve leaflets [75]. In this regard, it is interesting to note that the mitral valve in clinically healthy cKCs is significantly flatter (e.g. reduced leaflet tenting) than in healthy dogs from other investigated breeds [76], indicating that altered valve geometry could be a predisposing factor in this breed.

## Conclusions

In conclusion, we analysed eight dog breeds and identified an increased mutational load in the cKCs breed, which is highly bottlenecked. This breed has a high frequency of MMVD. Looking for derived alleles of high frequency specifically in this breed, we identified four highly conserved variants in protein coding genes that may relate to MMVD as well as several candidate regulatory variants near the *NEBL* gene. We find that some of these NEBL variants are associated with MMVD in dachshund and may be affecting gene expression in multiple relevant heart cell types and thus probably affect heart function. These results merit further validation in separate cohorts.

## Methods

### Ethics statement

All dog DNA was donated with the pet owners' consent. Blood and tissue from Swedish dogs were sampled under ethical approvals C12/15, C318/9 and C139/9 granted by "Uppsala djurförsöksetiska nämnd". Sampling of blood from Danish dachshunds was carried out prior to 2001 under a general Danish agreement allowing sampling in conjunction with clinical examinations.

### Sample collection and DNA extraction for whole genome sequencing

Blood from 20 unrelated individuals of each of the following eight dog breeds (amounting to a total of 160 individuals): beagle, cKCs, German shepherd, golden retriever, Labrador retriever, standard poodle, Rottweiler and WHwt; was collected and stored in EDTA tubes, with the purpose of achieving a representative sample of the genetic variation existing in these common dog breeds. All sampled dogs were privately owned and of Swedish origin with the exception of 10 German shepherds, 10 golden retrievers and 10 Labrador retrievers that were sampled in the US. DNA was extracted from blood using standard procedures.

## Sequencing, mapping and variant calling

Paired-end sequencing libraries were prepared using TruSeq DNA v2 sample preparation kit according to the manufacturer's protocol and sequenced in one run of paired-end sequencing with 100bp read length using a HiSeq2000/2500 system with v2 reagents (*Illumina Inc.*, *USA*). Raw sequence read data were deposited in the SRA under the following BioProject accession numbers: PRJNA693123. Sequencing reads were mapped to the dog reference genome (CanFam 3.1) using *BWA*–mem [77] and duplicated reads marked using *Picard* v1.92. Indel realigning, left aligning of indels and base recalibration was performed using GATK 2.7.2 according to the best practice recommendations [78]. Single nucleotide variants and indels were called using GATK 3.7 also following the recommendations in the GATK best practice workflow. Variants were called jointly on all 160 dogs and variant calibration was carried out using the following parameters for SNVs: "*-an MQRankSum -an ReadPosRankSum -an QD -an FS -an DP -an SOR -an MQ*", where approximately 4,000,000 SNVs detected in a previous whole genome sequencing analysis of dogs and wolves [79] were used as training set and nearly 100,000 SNVs on the Illumina 170 K HD array found to be segregating in a set of 500 dogs from 40 different breeds [80] were used as a truth set. Filtering was then performed to keep 99.9% of the SNVs in the truth set.

Similarly, variant calibration for indels was carried out using: "*-an MQRankSum -an ReadPosRankSum -an QD -an FS -an DP -an SOR*" and indels from Axelsson et al [79] used as both training and truth set. Filtering was performed to keep 99.0% of the indels in this truth set. Publicly available whole genome sequence data from an Andean fox [81] was used to infer ancestral states of dog variants. For this purpose, we genotyped the Andean fox data using *GATK hapotypecaller (-gt_mode GENOTYPE_GIVEN_ALLELES)* at all single nucleotide variant and indel sites segregating in the 160 dogs.

Large deletions (200 bp– 50Kb) were detected jointly on all 160 dogs using the *Genome-STRiP* (2.00.1710) [82] pipeline (*SVPreprocess*, *SVDiscovery* and *SVGenotyper*). Prior to deletion detection a reference meta data bundle for CanFam3.1 including the following files was created: a genome mask file (*org.broadinstitute.sv.apps.ComputeGenomeMask*), a reference.gc.mask file based on the repeat masked CanFam3.1 reference sequence and a canFam3.read-DepthMaskFile.bed including all autosomal sequence coordinates. *SVDiscovery* was run with the followoing parameters: *-minimumSize 100 -maximumSize 100000 -windowPadding 3000 -windowSize 5000000*. CNVs were also detected jointly on all dogs using the *CNVDisvorey* pipeline of *GenomeSTRiP* (2.00.1710). The pipeline was run with the following parameter settings: *-tilingWindowSize 5000 -tilingWindowOverlap 2500 -maximumReferenceGapLength 2500 -boundaryPrecision 200 -minimumRefinedLength 2500*. Redundant deletion and CNV calls and low-quality variants or genotype calls were filtered using *SVAnnotator* with the following parameter setting: *-A Redundancy -duplicateOverlapThreshold 0 -filterVariants TRUE -filterGenotypes TRUE*. Deletions and CNVs observed in a single individual were also removed from subsequent analysis.

## Variant detection power and genotype accuracy

Power, accuracy and false discovery rate (*FDR*) of the SNV calling and genotyping was estimated by comparing variant calls from our whole genome resequencing data with those based on array genotyping (HD 170 canine array) in the same set of 20 cKCs individuals sequenced here [80]. Power was calculated as the fraction of biallelic variable sites detected using array genotyping that was detected using whole genome resequencing. Similarly, genotype accuracy was estimated as the fraction of overlapping WGS calls showing identical genotypes in the array data. The *FDR* of the SNV discovery was estimated for sites interrogated using array

genotyping (i.e. present on the HD 170 canine array) and calculated as the fraction of these sites that were variable in our WGS data that were found not to be variable using array genotyping.

## Variant annotation

SNVs and indels were first annotated using *SNPeff* [83] and structural variants using custom written Perl scripts available at https://github.com/erik-axelsson/delvar_MMVD/ (this repository contains all scripts used for all bioinformatic analyses described in this article). We then used evolutionary constraint to identify variants that are likely to be functional. For this purpose, we downloaded *PhyloP* scores (http://genome.ucsc.edu) estimated for the human reference genome based on comparisons of 46 mammals and 100 vertebrates, respectively. To assign *PhyloP* scores to dog variants we then used *LiftOver* to find the orthologous location of dog variants (CanFam3.1) in the human reference assembly (hg38) and then mapped these positions back to CanFam3.1 to confirm reciprocal mapping. This way mammalian and vertebrate *PhyloP* scores were assigned to a total of 8,419,950 dog SNVs and indels. Similarly, *phastcon* elements already mapped to CanFam3.1 were downloaded from the UCSC website and used to annotate structural variants with potential functional significance.

## Accumulation of deleterious mutations

We compared the relative abundance of potentially deleterious variants in all possible breed pairs to test if the varying intensities of recent breed bottlenecks have affected the rate of accumulation of deleterious genetic variation in dog breeds. To this end we use the $R_{A/B}$ statistics [3] to compare the relative abundance of derived potentially deleterious alleles, at all biallelic SNVs and INDELs, in different breed pairs. Similarly, as also described in Do et al [3], we then calculate the relative probability, $R^2_{A/B}$, that a breed is homozygous for derived alleles whereas the other breed is not. We assessed the variance in the relative abundance of deleterious variation for breed pairs using Weighted Block Jackknife, whereby the dog genome was split into 50 contiguous windows followed by recalculation of $R_{A/B}$ leaving out one window at a time. Significance was then estimated based on the number of standard errors away from the expectation $R_{A/B} = 1$ and p-values calculated using Z-scores assuming a normal distribution. P-values were Bonferroni-corrected to correct for multiple testing. We estimate $R_{A/B}$ for different functional categories of sites, including those annotated as *synonymous coding* and are likely to have no effect on fitness, and two sets of potentially functional sites (*nonsynonymous coding* and *loss of function* (*LoF*)) and for two categories of sites evolving under different evolutionary constraint as assessed based on the *PhyloP* statistic (*pp*) calculated in alignments of 100 vertebrates; i.e. *moderately conserved sites* ($> = 2\ pp <5$) and *highly conserved sites* ($pp > = 5$). *LoF* included variants affecting sites characterized as belonging to one of the following *SNPeff* categories: *CODON_CHANGE_PLUS_CODON_DELETION', 'CODON_CHANGE_PLUS_CODON_INSERTION', 'CODON_DELETION', 'CODON_INSERTION', 'EXON_DELETED', 'FRAME_SHIFT', 'SPLICE_SITE_ACCEPTOR', 'SPLICE_SITE_DONOR', 'START_LOST', 'STOP_GAINED'*.

To specifically test if hitchhiking may have contributed to the accumulation of deleterious genetic variation during cKCs breed formation we used *Sweepfinder 2* [84], with grid size of 10K (option -lg 10000) on a pre-computed empirical frequency spectrum (generated with the option -f), to identify regions in the cKCs genome that may have been targeted by recent positive selection. 10 kb windows with the top 5% highest composite likelihood ratio scores were chosen to represent putative targets of selection. Neighbouring windows under putative selection were concatenated and adjacent (within 100 kb) concatenated regions aggregated,

resulting in a total of 88 selection regions spanning 129 Mb across the entire genome (Table 27 in S1 Text). We then masked these regions from the cKCs genome and re-estimated $R_{A/B}$ for highly conserved sites to measure the relative abundance of deleterious alleles outside of regions affected by hitchhiking.

## Demographic analyses

A neighbor-joining tree based on genome-wide pair-wise $F_{ST}$ estimates was drawn using *SplitsTree* [85]. Heterozygosity was calculated per breed at all biallelic sites found to be segregating in the entire data set. Runs of homozygosity (ROH) in each breed were detected using PLINK [86] following the commands used in Freedman et al [6]: "*plink—tfile INFILENAME—homozyg—homozyg-snp 200—homozyg-kb 2000—homozyg- window-missing 100—homozyg-window-het 10—allow-no-sex—dog—out OUTFILENAME*". Similarly, decay of linkage disequilibrium in each breed was estimated for chromosome 1 using PLINK and the following command: "*plink—vcf INFILENAME—double-id—dog snps-only—maf 0.15—r2—ld-window 99999—ld-window-r2 0—out OUTFILENAME*". The demographic histories of dog breeds were inferred based on breed specific site frequency spectra by the software *epos* [87] using the greedy option and a mutation rate of $4.5 \times 10^{-9}$ per base pair and generation [88]. The site frequency spectra were obtained from all sites in the genome that were sequenced in all 20 individuals from each breed and confidence intervals of estimated population sizes were estimated using 1000 bootstrap replicates of these frequency spectra.

## Identification of candidate risk variants for MMVD

With the aim of utilizing the data generated here to explore the molecular mechanisms underlying MMVD we hypothesized that one or several high frequency derived alleles that are rare, or absent, in the other breeds sequenced here may predispose cKCs to developing MMVD. We also reasoned that potential disease-causing variants are likely to affect evolutionarily conserved sites and that alleles contributing to increased risk of developing MMVD may preferentially be located near, or within, genes that are important for proper heart function and/or of relevance to MMVD pathology. To identify high frequency derived alleles we used Weir and Cockerham's estimator [89] to calculate $F_{ST}$ for all SNVs (including multi-allelic) and indels across all possible breed pairs. Differentiation in one breed relative to all other breeds was calculated as the mean of all pair-wise estimates (average pair wise $F_{ST}$) and a cut-off at average pair-wise $F_{ST} >= 0.7$ was used to characterize variants as highly differentiated in a particular breed. Highly differentiated structural variants were identified using the same approach but population differentiation was measured using the $V_{ST}$ statistics ($V_{ST} >= 0.7$), rather than $F_{ST}$ [90].

Evolutionary conservation was evaluated using the *PhyloP*-statistics for SNVs and indels, whereas overlap with a positive *PhastCons* lod-score was used to identify structural variants affecting conserved sequence. To prioritize potential MMVD risk variants among the high frequency derived variants detected in cKCs, we first identified variants that affect *highly conserved sites* ($PhyloP_{100vertebrates} > 5$, n = 50) located within 5 Kb up- and downstream of genes with known function of potential relevance to MMVD pathology. Gene function was interrogated using gene-ontology annotations and literature searches. Second, we also prioritized potential MMVD-risk variants by identifying high frequency derived variants in cKCs that affect moderately or highly conserved sites ($PhyloP_{100vertebrates}$ or $PhyloP_{46mammals} >= 2$) located within 5 Kb up- and downstream of genes (n = 269) that were previously shown to be downregulated in canine MMVD cases relative to in healthy controls [37]. A total of 10 SNVs were characterized as candidate MMVD-risk variants based on these prioritizations.

## Within breed association analyses for MMVD

To explore if any of the candidate variants identified in cKCs are likely to contribute to the risk of developing MMVD, we investigated whether genotypes at these variants may be used to predict MMVD in individual dachshunds. The dachshund breed was selected for this purpose because, similar to in cKCs, MMVD is heritable in dachshunds [20], but the prevalence of MMVD in this breed is moderate arguing that potential risk alleles that have become fixed in cKCs may be segregating and therefore detectable by association analysis in dachshund. One hundred and twenty-two (122) privately owned Swedish dachshunds aged >4.8 years were sampled with the owners' consent from a network of Swedish breed clubs. To assess cardiac health status of the dachshunds, dogs underwent cardiac auscultation in a quiet examination room at SLU, during which the presence of a systolic heart murmur was noted and graded on a scale from 1 to 6 in accordance with established guidelines [91]. As inclusion criteria for the study, dogs had to either have evidence of MMVD or be free from physical or echocardiographic evidence of cardiac disease. Dogs with congenital heart disease or other acquired cardiovascular disorders were not included in the study. Dogs in need of heart failure therapy were allowed into the study. Echocardiography using an ultrasonographic unit equipped with a 5–1 MHz transducer and ECG monitoring was then performed to verify a diagnosis of MMVD and to exclude other primary or secondary cardiac diseases. Diagnosis of MMVD was based on the presence of characteristic valvular lesions of the mitral valve apparatus (thickened and prolapsing mitral valve leaflets) and detection of mitral regurgitation [92]. MMVD severity was determined on the basis of the American College of Veterinary Internal Medicine (ACVIM) consensus statement classification system for MMVD [41]: Stage A; dogs that lacked identifiable structural signs of MMVD on the echocardiogram, stage B1; preclinical dogs with characteristic signs of MMVD on the echocardiogram, but without significant evidence of left-sided cardiac enlargement (i.e. left atrial to aortic dimension (LA/Ao) < 1.6, and left ventricular end diastolic diameter normalized for body weight (LVIDdN) < 1.7) in response to the disease [93], stage B2; preclinical dogs with characteristic signs of MMVD on the echocardiogram and evidence of left-sided cardiac enlargement (i.e. LA/Ao $\geq$ 1.6, and LVIDdN $\geq$ 1.7) in response to the disease, stage C; dogs with current or previous (stabilized on medical treatment) clinical signs of CHF (such as cough, dyspnea/tachypnea, nocturnal restlessness and/or exercise intolerance), and echocardiographic and/or radiographic changes compatible with CHF. At the visit to the clinic, a blood sample was collected in EDTA tubes from which DNA was extracted following standard procedures. Candidate markers were genotyped using either PCR and Sanger sequencing or TaqMan assays (Thermofisher, Massachusetts) (see Table 28 in S1 Text for primer sequences and TaqMan target sequences). PCR reactions were prepared using AmpliTaq Gold DNA polymerase from Applied Biosystems (Massachusetts, USA) according to manufacturer's instructions at a final volume of 20 ul. The PCR reaction was carried out in an Applied Biosystems 2720 Thermal Cycler (Massachusetts, USA) with a program of 95 C (10 min) initial activation followed by 35 cycles of 95 C (15 s.) denaturation, XX C (30 s.) annealing (See Table 28 in S1 Text for individual annealing temperatures) and 72 C (1 min.) elongation. The program ended with 72 C (5 min) final elongation. PCR products were purified using Exo I and FastAP from Thermofisher (Massachusetts, USA) and sent for sequencing to Eurofins (Luxembourg).

We used linear regression as implemented in *R* (RStudio version 1.1.453) to first test if age at examination (*age*), sex (*sex*) or candidate variant genotype (*genotype*), respectively, can predict binary disease status (*status*, healthy or affected) in dachshunds (univariate models: *(lm (status~age)), (lm(status~sex))* and *(lm(status~genotype)))*. Given that MMVD has an age dependent debut and that the disease progresses with increasing age, we next evaluated if age

at examination (*age*), candidate variant genotype (*genotype)* and *interactions* between these terms can predict graded disease status (*graded_status*), rather than binary disease status, using the following model: *lm(graded_status~genotype\*age)*. Furthermore, given that previous analyses have documented an earlier disease debut in male dachshunds [20] we also assessed whether a model that also included sex (*sex*), (*lm(graded_status~genotype\*age\*sex))*, could predict graded disease status. Finally, to maximize discovery power for a late onset, polygenic disease such as MMVD, we also used linear regression, using the following model: *(lm (status~genotype))*, to test for a direct association between genotype and binary health status (affected or healthy) in subsets of the dachshund population that contained only young cases and old controls. For this purpose, we created four subsamples characterized by including increasingly young cases (aged under 10, 9, 8 or 7 years at examination, respectively) and increasingly old controls (aged more than 8, 9, 10 or 11 years at examination, respectively).

In addition to in the dachshund population, we also explored whether candidate variant genotypes could predict presence of MMVD in a small beagle population used for mainly educational purposes at SLU. We followed the same procedures as described above for dachshunds to examine (for signs of MMVD), sample (for blood) and genotype (the 10 candidate markers) all individuals aged >3 years (n = 22) from this population. Statistical analyses of the beagle data were carried out as described above for dachshunds.

Haplotypes and decay of linkage disequilibrium for the six *NEBL* variants studied here were inferred in the dachshund population using *Fastphase* [94] and *Haploview* [95], respectively.

## Tests for association between candidate variants and gene expression

Six of the variants analysed here are located near or within *NEBL*, a gene that was previously identified as having reduced gene expression in heart tissue collected from a small number of dogs affected by MMVD relative to in healthy individuals [37]. To test if the *NEBL* SNVs analysed here may be associated with altered expression of *NEBL*, or other nearby genes, we used quantitative PCR (qPCR) to measure RNA levels in tissue samples from mitral valve (n = 23), papillary muscle (n = 23) and left ventricular heart wall (n = 21), from a total of 23 dogs representing eight different breeds (Table 23 in S1 Text). Tissue sampling was done with the dog owners' consent within 30 minutes *postmortem* from dogs that were euthanized for other reasons than this study. Upon sample collection, tissue samples were transferred to Tissue-Tek O. C.T. (*Sakura Finetek*, *Japan*) and then snap frozen in liquid nitrogen (mitral valve) or just snap frozen (Thermofisher, Massachusetts) (papillary muscle and left ventricular heart wall) and stored at -80 C until analysis. RNA was extracted from the tissue samples using the *Qiagen All-Prep DNA/RNA/miRNA kit* (Qiagen, Germany) and RNA integrity was assessed using the *Agilent TapeStation system* (Agilent, California). cDNA was synthesized from the RNA using the Advantage RT-for-PCR kit (Takara Bio, California). For the qPCR we designed primers (Table 29 in S1 Text) targeting four genes and five transcripts near the analysed variants (*NEBL* isoform nebulette, *NEBL* isoform LIM-nebulette, *MLLT10*, *SKID1A* and *C10orf113*) as well as two housekeeping genes used for normalization (*RPL13A* and *G3P*). For each *NEBL* transcript two sets of primers were designed to allow for replication of test results. SYBR Green (Thermofisher, Massachusetts) was used to label cDNA and reactions were set up according to the manufacturers protocol at a final reaction volume of 5 ul. Reactions were run in a *QuantStudio 6 Flex Real-Time PCR system* (Thermofisher, Massachusetts) with a program of 95 C (10 min) initial activation followed by 40 cycles of 95 C (15 s.) denaturation and 60 C (1 min,) elongation. In a subset of healthy heart tissues, we first evaluated if the five transcripts of interest are expressed in heart. We detected reliable heart expression (defined as C(T)<30 in undiluted cDNA samples from any of the three heart tissues) for *NEBL* isoform nebulette,

*NEBL* isoform LIM-nebulette and *MLLT10* and therefore only continued to evaluate expression of these transcripts in the complete sample. Before analysing the complete sample, we established optimal cDNA starting concentrations and confirmed qPCR efficiency using dilution series in a single cDNA sample from each tissue. Relative expression levels for the five transcripts were then quantified across all samples using the 2(-Delta Delta C(T)) method [96], for which normalization was performed using both reference genes. C(T) values were always measured using four technical replicates. To evaluate if gene expression of any of the three transcripts may be influenced by the *NEBL* candidate variants studied here we first used PCR and Sanger sequencing (as described above for dachshunds) to genotype the six SNVs in all 23 donor dogs. Next, we used linear regression in R (*lm(relative_expression~genotype))* to test for associations between relative expression levels and genotype assuming an additive model. For our main regression analysis, we included data from all dogs for which sample RNA integrity exceeded six (RIN-value > 6) [97] and the technical replicate standard deviation did not exceed 0.3 (s.d. < 0.3). We also studied expression in five additional subsamples of the data representing different combinations of inclusion criteria (*i)* RIN>5, s.d.<0.3, all dogs included; *ii)* RIN>7, s.d.<0.5, all dogs included; *iii)* RIN>5, s.d.<0.5, four Grand danois affected by dilated cardiomyopathy excluded *iv)* RIN>6, s.d.<0.3, four grand danois affected by dilated cardiomyopathy excluded *v)* RIN>7, s.d.<0.5, four grand danois affected by dilated cardiomyopathy excluded; Table 24 in S1 Text) and primarily relied on results that replicated in all six subsamples.

## Electrophorectic mobility shift assays

As a first means to investigate a potential regulatory role for SNVs associated with MMVD-risk in this study we used EMSA to assess if variants *NEBL* 1, 2, 3 and 4 bind protein in relevant cell types and whether protein binding differs between alleles at these sites. For this purpose, we designed DNA oligos (with and without biotin labelling) spanning 40 nucleotides centred around each allele of the four SNVs (Table 30 in S1 Text). For *NEBL* 1 we designed additional oligos to test if a non-conserved SNV (named *NEBL* 1.2) located 10 bp downstream of *NEBL* 1 may interfere with protein binding. Specifically, we designed oligos containing the alternative allele of *NEBL* 1.2 and either *NEBL* 1 reference, or alternative alleles, respectively (Table 30 in S1 Text). Protein binding to all oligos was assayed in nuclear extracts from three different cell types; *i)* the MDCK cell line is derived from dog (Cocker spaniel) kidney tubule epithelia and was used as a representative of a non-cardiac derived cell line that allowed us to test binding in a setting where both protein and probe sequence originate from dog; *ii)* In absence of immortalized cell lines derived from mitral valve, we used RAVIC cells as a model for heart valve interstitial cell biology. RAVIC cells derive from rat (Sprague Dawley rats) aortic valvular interstitial cells [47]. Similar to the mitral valve, the aortic valve is located in the left side of the heart. Both valve types are composed of an outer layer of endothelial cells and a matrix of collagen, elastic fibers, proteoglycans and glycoproteins that is populated by the valvular interstitial cells (VICs); *iii)* H9C2 cells are derived from embryonic ventricular BDIX rat heart tissue [98]. H9C2 normally have a skeletal muscle like phenotype but can be induced to differentiate into cardiomyocytes if treated with retinoic acid (RA) [99]. Here we used RA treated H9C2 cells to assay binding in cardiomyocytes.

MDCK and RAVIC cells were seeded in culture flasks containing DMEM-F12 (Thermo-fisher, California) supplemented with 10% fetal bovine serum as well as penicillin and streptavidin. The cells were grown at 37˚C in atmosphere containing 5% $CO_2$ until reaching confluency. Cell medium was changed every 2 days. H9C2 cells were grown in special DMEM containing 1500mg/L of sodium bicarbonate (ATCC 30–2002) (ATCC, Virginia) but

otherwise treated as described above initially. Before reaching confluency the H9C2 cells were differentiated by RA treatment for five consecutive days as described in [99]. Nuclear protein was extracted using the *NucBuster Protein Extraction kit* (Novagen, Germany) and EMSAs were performed using the *LightShift Chemiluminescent EMSA kit* (ThermoFisher, Massachusetts). EMSA reactions were loaded onto 5% Criterion TBE Polyacrylamide Gel (Bio-Rad, California) and transferred to Genescreen PLUSR Hybridization Transfer Membranes (PerkinElmer, Massachusetts).

## Luciferase assay

Based on the results of the EMSAs we decided to use luciferase assays to further evaluate the regulatory potential of candidate variant *NEBL* 1 in the two heart cell lines described above (RAVIC and H9C2). Two long fragments (473 bp) spanning *NEBL* 1 and 1.2 were amplified (see Table 31 in S1 Text for primers) from DNA isolated in a dog carrying reference alleles, and a cKCs carrying alternative alleles, at both sites, respectively. Two pGL4.26 (Promega, Wisconsin) vector constructs carrying the luciferase reporter gene *luc2* and fragments with either reference or alternative allele combinations were generated by cutting both vector and fragment using restriction enzymes EcoRV and HindIII followed by ligation of fragment and vector using T4 ligase. Successful ligation was verified using Sanger sequencing (Eurofins, Luxembourg). *One shot top 10 chemically competent cells* (Invitrogen, California) were then transformed with pRL-TK control vectors containing Renilla luciferase and either one of the two vector constructs or empty pGL4.26 vector. Transformed cells were plated and grown on ampicillin containing LA-plates overnight to select for successful transformation. Large vector quantities were produced by transferring a few colonies from each transformation to 100 ml ampicillin supplemented LB medium for overnight growth. Vector DNA was then purified from the bacterial clones using Qia prep spin Miniprep kit (Qiagen, Germany). Prior to transformation, heart cells (RAVIC and H9C2) were first grown in T75 flasks according to the protocol described above. Subsequently, approximately 200,000 cells were seeded into each well on two 6-well plates and allowed to grow until 100% confluence for RAVIC cells and 60–70% confluency for H9C2 cells. H9C2 cells were then differentiated to cardiomyocytes using RA as described above. Transfection of heart cells was performed using *jetPRIME* (Polyplus, France) according to the manufacturer's recommendations, but using 1 ug of vector DNA for H9C2 cells, as indicated by our transfection optimization experiment. Each vector combination was transfected in four replicates, except for the empty pGL4.26, for which three replicates were used. Relative luciferase activity was quantified using the *Dual-Luciferase Reporter Assay System* (Promega, Wisconsin) on an *Infinite 200 PRO plate reader* (Tecan, Switzerland).

## Across breed association analysis for MMVD

As a further means of exploring if any of the candidate MMVD variants analysed in this study may affect the risk of developing MMVD in other breeds than dachshunds, we tested if average breed-specific allele frequencies at candidate variants could predict incidence of MMVD across a small set of dog breeds. For this purpose, we genotyped all candidate variants in cocker spaniel (n = 25), Norfolk terrier (n = 11), Norwich terrier (n = 11), Grand Danois (n = 10) and Dobermann pinscher (n = 9), all of which were sampled in Sweden. In addition to the eight breeds for which allele frequencies could be calculated based on the whole genome sequence data generated here, and the dachshunds (n = 122) genotyped for the within-breed association analysis, allele frequencies were hence calculated in a total of 14 breeds. To estimate MMVD incidence for these breeds in Sweden we studied MMVD-related claims from the insurance company *AGRIA Pet Insurance* during the time period 2011–2016. Approximately

38% of the Swedish dog population is insured by Agria. Owners of dogs insured for veterinary care were reimbursed for claims if the total cost of all veterinary appointments during a 125-day period exceeded the deductible of the insurance. Incidence rate per breed was calculated as the number of MMVD-related claims from the veterinary care insurance per 10,000 dog-years-at-risk (DYAR). Two incidence rates were calculated; *i)* an MMVD specific incidence rate based on claims involving MMVD specific diagnostic codes (CA222—MMVD, CA2221- mitral MMVD, CA2222—tricuspid MMVD, CA2223—rupture of chordae tendinae, CA2224—rupture of left ventricle) and *ii)* a more general, yet MMVD focused, heart disease incidence rate based on the MMVD specific claims described above as well as a set of general diagnostic codes for heart disease (CA01-signs of unspecified heart disease, CA011-heart murmurs of unspecified cause, CA012-signs of cardiomegaly, CA013-signs of congestive heart failure). Only the first MMVD-related (or heart disease related) claim for each dog was counted when estimating incidence. The diagnostic codes used in the database are based on the diagnostic registry developed by the Swedish Association of Veterinary Clinics and Hospitals, and assigned by the examining veterinarians. Dog-years at risk (DYAR) for the veterinary care insurance were calculated from the start of the insurance (if insured at/after January 1, 2011) or start of observation period (if insured before January 1, 2011) to the first MMVD-related claim or to withdrawal from insurance/the end of observation period (December 31, 2016) in dogs without MMVD-related claims. All incidence analyses were performed in RStudio version 1.2.1335. Linear regression, as implemented in R *(lm(incidence~allel frequency))*, was used to test if candidate variant allele frequencies can predict MMVD-incidence.

## Supporting information

**S1 Text. Supporting figures and tables.** This file contains all supporting figures and tables referred to in the manuscript.
(DOCX)

**S1 Data. Large deletions detected in sequenced dogs.** Location (*chr*, *start*, *end*) and size (*length* in bp) of large deletions detected in the 160 dog genomes and found to be segregating in at least two of the sequenced individuals. For deletions overlapping genes, gene name (*gene_name*), ensemble gene id (*gene*), ensemble transcript id (*transcript*), gene description (*description*) and gene ontology terms (*GO-annotations*) associated with the gene are reported. deletions overlapping conserved elements are flagged as conserved (*phastcon*) and the highest phastcon lod score within the boundaries of the deletions is reported (*phastcon_lod*). Average allele frequency of reference (*breed_name_A*) and deletion (*breed_name_B*) allele, total number of chromosomes sampled per breed (*breed_name_sz*) and average pairwise breed differentiation for the deletion (*breed_name_fst*) is reported for each breed. Total differentiation across the entire sample (total_FST) is also reported for each deletion.
(TXT)

**S2 Data. Large copy number variants detected in sequenced dogs.** Location (*chr*, *start*, *end*) and size (*length* in bp) of CNVs detected in the 160 dog genomes and found to be segregating in at least two of the sequenced individuals. For CNVs overlapping genes, gene name (*gene_name*), ensemble gene id (*gene*), ensemble transcript id (*transcript*), gene description (*description*) and gene ontology terms (*GO-annotations*) associated with the gene are reported. CNVs overlapping conserved elements are flagged as conserved (*phastcon*) and the highest phastcon lod score within the boundaries of the CNV is reported (*phastcon_lod*). Average diploid copy number (*bread_name_CNV*), range of diploid copy numbers (*range_bread_name*), where measures in each individual is separated by "_", and average pairwise breed differentiation for

the CNV (*breed_name_VST*) is reported for each breed. Total differentiation across the entire sample (total_VST) is also reported for each CNV. *ANOVA* reports p-values from anova analysis testing if average copy numbers deviate between breeds.
(TXT)

**S3 Data. Phenotypes and candidate marker genotype for the beagle and the two dachshund populations studied.** *Id*, *breed* (dachshund or beagle), *nationality* (Swedish or Danish), *sex* (0-femal, 1-male), *MMVD_status* (0-healthy, 1-affected), *graded_MMVD_status* (according to ACVIM guidelines: 0-A, 1-B1, 2-B2, 3-C), *age* (age at examination) and genotypes for the 10 candidate markers (NEBL 1, 2, 3, 4, 5, 6, SORBS2, LPHN2, HDGFL1 and HTR1F) in the dachshunds and beagle used for within breed association analyses.
(TXT)

## Acknowledgments

We thank Lucy Lin Cui for kindly providing the RAVIC cells and Karin Hultin Jäderlund and Katja Höglund for providing dog samples. We thank *Agria Pet Insurance* for granting access to insurance data for estimating MMVD incidence. The authors would like to acknowledge support from Science for Life Laboratory, the National Genomics Infrastructure and NGI-Uppsala SNP&SEQ for providing assistance in massive parallel sequencing and computational infrastructure. The computations were performed on resources provided by SNIC through Uppsala Multidisciplinary Center for Advanced Computational Science (UPPMAX) under project b2013119.

## Author Contributions

**Conceptualization:** Erik Axelsson, Ingrid Ljungvall, Priyasma Bhoumik, Åke Hedhammar, Philippe Gruet, Jens Häggström, Kerstin Lindblad-Toh.

**Data curation:** Erik Axelsson, Karolina Engdahl.

**Formal analysis:** Erik Axelsson, Kerstin Lindblad-Toh.

**Funding acquisition:** Erik Axelsson, Ingrid Ljungvall, Ragnvi Hagman, Kerstin Lindblad-Toh.

**Investigation:** Erik Axelsson, Laura Bas Conn, Eva Muren, Åsa Ohlsson, Karolina Engdahl, Dmytro Kryvokhyzha.

**Methodology:** Mats Pettersson.

**Project administration:** Erik Axelsson, Priyasma Bhoumik.

**Resources:** Ingrid Ljungvall, Åsa Ohlsson, Lisbeth Høier Olsen, Ragnvi Hagman, Jeanette Hanson, Jens Häggström.

**Software:** Erik Axelsson, Priyasma Bhoumik, Dmytro Kryvokhyzha, Olivier Grenet, Jonathan Moggs, Alberto Del Rio-Espinola, Christian Epe, Bruce Taillon, Nilesh Tawari, Shrinivas Mane, Troy Hawkins.

**Supervision:** Erik Axelsson, Priyasma Bhoumik, Mats Pettersson, Olivier Grenet, Jonathan Moggs, Alberto Del Rio-Espinola, Bruce Taillon, Nilesh Tawari, Troy Hawkins, Åke Hedhammar, Philippe Gruet, Jens Häggström, Kerstin Lindblad-Toh.

**Validation:** Erik Axelsson.

**Visualization:** Erik Axelsson.

**Writing – original draft:** Erik Axelsson, Kerstin Lindblad-Toh.

**Writing – review & editing:** Erik Axelsson, Ingrid Ljungvall, Priyasma Bhoumik, Laura Bas Conn, Eva Muren, Åsa Ohlsson, Lisbeth Høier Olsen, Karolina Engdahl, Ragnvi Hagman, Jeanette Hanson, Dmytro Kryvokhyzha, Mats Pettersson, Olivier Grenet, Jonathan Moggs, Alberto Del Rio-Espinola, Christian Epe, Bruce Taillon, Nilesh Tawari, Shrinivas Mane, Troy Hawkins, Åke Hedhammar, Philippe Gruet, Jens Häggström, Kerstin Lindblad-Toh.

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
