## [Decision Letter · Decision Letter 0]

28 Apr 2021

Dear Dr Axelsson,

Thank you very much for submitting your Research Article entitled 'The genetic consequences of dog breed formation - accumulation of deleterious genetic variation and fixation of mutations associated with myxomatous mitral valve disease in cavalier King Charles spaniels' to PLOS Genetics.

The manuscript was fully evaluated at the editorial level and by independent peer reviewers. The reviewers appreciated the attention to an important topic but identified some concerns that we ask you address in a revised manuscript

We therefore ask you to modify the manuscript according to the review recommendations. Your revisions should address the specific points made by each reviewer.

[LINK]

Yours sincerely,

Gregory Copenhaver

Editor-in-Chief

PLOS Genetics

Gregory Barsh

Editor-in-Chief

PLOS Genetics

Reviewer's Responses to Questions

**Comments to the Authors:**

Reviewer #1: The manuscript entitled « The genetic consequences of dog breed formation - accumulation of deleterious genetic variation and fixation of mutations associated with myxomatous mitral valve disease in cavalier King Charles spaniels » by Erik Axelsson  et al analyzed eight dog breeds by WGS (n=20 dogs in each breed) and identified an increased mutational load in one breed the King Charles spaniels (cKCs) breed. Then they investigate the myxomatous mitral valve disease (MMVD) highly present in this breed. The authors performed an approach looking for derived alleles of high frequency specifically in the breed. They identified highly conserved variants in protein coding genes (n=4) that may relate to MMVD and propose several candidate variants near the NEBL gene. Then they find that some of these NEBL variants are associated with MMVD in dachshund breed and may be affecting gene expression in multiple relevant heart cell types and thus probably affect heart function. They conclude saying that these results merit further validation in separate cohorts.

• What are the genetic relationships of the eight breeds, do they belong to the same dog clade, to different clades ? It would interesting to present this to readers for two reasons (1) general information about these 8 breeds and (2) do cavalier King Charles spaniels are outlier of all others or cluster within a subgroup ?

• "To identify variants linked to this disease we next characterize mutations that are common in cKCs, but rare in other breeds, and then investigate if these mutations can predict MMVD in dachshunds". Why specifically dachshunds ? I understand because the risk alleles segregate at intermediate frequencies while seem to be fixed in cKCs ? Is that all ? Are there other breeds with this heart condition with good clinical data?

• "breed dogs carry 2-3% more derived, potentially deleterious, alleles at evolutionarily conserved amino acid changing sites than wolves". How is defined a derived allele ? I image several ways - postulate that ancestral allele is from a wild canid, from the wolf ? or using dog variant calling the REF allele is assumed as the ancestral allele and ALT allele the derived?. Well I think I understand the ALT allele is used when the authors use the statistic that measures the number of derived mutations observed in breed A that are not observed in breed B, and then calculates the ratio, RA/B, of the two numbers. Maybe the authors should provide some explanations of the derived allele, this could be clarified.

• "we calculated average pairwise FST for all variants across all breeds to identify high frequency derived alleles". Why not a across-breeds chi-sq test or fisher test ?

• page 18 : therefor > therefore

• How do you understand that accelerated accumulation of mutational load during breed formation is seen only in cKCs at least for the additive model for allelic effects, ("cKCs carry 6.4% to 12.5% more derived mutations at highly conserved sites relative to the other breeds") ? No accelerated accumulation of mutational load is observed in other breeds because of different breeding practices ? because of demographic effect during/after breed formation ? and would that mean that the method to use mutational load to investigate a disease can not be transferred to many dog breeds ? to many diseases ?

It would be nice to discuss how to transfer this approach (genetic load in breeds assuming additive or recessive model for allelic effects), into a more general approach to investigate disease when using the dog model and WGS data.

• Dachshund association analysis. This paragraph (p18) is more about the statistics results for 10 selected candidate variants, rather than an association analysis. I suggest to change the paragraph title.

• Across-breed association test : In this paragraph, you test if allele frequencies of the candidate variants can predict the incidence of MMVD. Is that done within each breed ? I suspect not. All breeds are used together for a linear regression test between genotype and the disease status ? Could you make that clearer.

Reviewer #2: Thank you for the opportunity to review this manuscript. This is a very well-written, interesting article and the analyses are comprehensive.The authors based on WGS show highest increase in derived alleles per cavalier King Charles spaniels genome compared to eight other breeds they studied. The authors used evolutionarily conserved bases in vertebrates to show the accumulation of deleterious mutations in cKCs. They also used the WGS data to identify highly conserved regulatory variants and other protein coding variants that could be linked to myxomatous mitral valve disease development in cKCs and daschunds. The authors have done a good job explaining the main limitations of the work and outlining why findings should be interpreted with caution. I found this to be a well-written, engaging, and thoughtful manuscript with important contributions to the use of SNPs in studies of dog breed characteristics.

The only major thing is maybe, the authors have done extensive work and would be good make the programming scripts available, so with the raw sequence data made available, maybe some part of the work can be considered reproducible.

MINOR CORRECTIONS

* Sequencing and variant discovery  The authors need to explain what is FDR in the methods.

* MMVD coding candidate variants

The citation for genes with known function of potential relevance to MMVD pathology

Reviewer #3: Comments to the authors

The authors have generated an impressive amount of data for this study. Overall I think there are some very interesting findings and I find no big flaws in the analyses. I am not a cell biologist, so I hope that one of the other reviewers can provide insight on the gene specific parts of the manuscript. I do have some general comments that I hope the authors can use.

First of all a lot of information is included in this manuscript. I think the manuscript could be improved if it gets shortened. For example much of the information presented in the Introduction and Results are repeated in the Discussion and Methods. By vetting out the repetitions the manuscript would get easier to read.

A bit more background on dog domestication demographics would be good. Page 5 L9-19 mentions the reduction in Ne during domestication but just the fold reduction from wolf to dog. It would give the reader a better idea for the Ne for the breeds included in present study if you include the estimated values for each breed as well.

Its mentioned several times that the cKCs have been through a more intense bottleneck(s) than the other breeds. Please include a reference for this or show it with your results i.e., heterozygosity, ROH. It also makes the reader wonder if some of the breeds have been through consecutive bottlenecks. Given the large literature on dog domestication this information is likely obtainable.

In the Discussion the authors speculate on the effect of past bottlenecks on the cKC breed (P27 L13-15). This could potentially be resolved by running some more demographic analyses.

There seem to be two different stories merged into one. First one is the cKCs part and second the Dachshund part. It makes sense to include them both as the authors are searching for variants associated with MMVD, but I suggest that the authors make a better bridge between the two parts. For example flesh out more clearly why Dachshunds were chosen (I guess because the dataset were available?). Maybe you could move the sentence “The Dachshund breed was….” (P43 L6-9) to the Introduction?

Minor comments

P9 L4: Is the dog genome chromosome level?

P37 L24: the GATK settings for the different parameters are missing.

L6: GATK parameters not provided.

P39 L3-10: This power analysis is a good inclusion.

P39 L14: will these custom scripts be made available?

P41 L18: Is LD information available for canines?

P44 L23-25: age, sex and genotype is included in the linear regression. I wonder if other traits like body mass, fitness etc. could influence the onset of MMVD, and if so the authors also tested these in the linear reg?

P45 L18-19: This sounds like it’s a laboratory line of dogs. If so is it selected to have higher expression of MMVD?

Figure 2e: it looks like a label is missing on this figure. The left part is an undescribed smear.

**Have all data underlying the figures and results presented in the manuscript been provided?**

Reviewer #1: Yes

Reviewer #2: Yes

Reviewer #3: Yes

PLOS authors have the option to publish the peer review history of their article (what does this mean?). If published, this will include your full peer review and any attached files.

Reviewer #1: No

Reviewer #2: No

Reviewer #3: No

---

## [Decision Letter · Decision Letter 1]

20 Jul 2021

Dear Dr Axelsson,

We are pleased to inform you that your manuscript entitled "The genetic consequences of dog breed formation - accumulation of deleterious genetic variation and fixation of mutations associated with myxomatous mitral valve disease in cavalier King Charles spaniels" has been editorially accepted for publication in PLOS Genetics. Congratulations!

The revised manuscript was seen by reviewers #1 and #3 of the original manuscript; as you will see, they are both positive.

Yours sincerely,

Gregory S. Barsh

Editor-in-Chief

PLOS Genetics

Gregory Copenhaver

Editor-in-Chief

PLOS Genetics

Comments from the reviewers (if applicable):

Reviewer's Responses to Questions

**Comments to the Authors:**

Reviewer #1: The revised version provides an improvement of the initial version. The authors modified, clarified and added information upon requests suggested. The revised version provides corrections, clarifies some technical aspects, the linear regression analyses, the Fst approaches, and presents a simple but clear phylogenetic tree. The authors have updated the Methods section. Typos have been corrected.

To this point, I have no major comments on this revised version.

Reviewer #3: The authors have carefully addressed all the comments I had in my previous review and I have no further changes to add. Well done.

**Have all data underlying the figures and results presented in the manuscript been provided?**

Reviewer #1: Yes

Reviewer #3: Yes

PLOS authors have the option to publish the peer review history of their article (what does this mean?). If published, this will include your full peer review and any attached files.

Reviewer #1: No

Reviewer #3: No

**Data Deposition**

http://datadryad.org/submit?journalID=pgenetics&manu=PGENETICS-D-21-00373R1

**Press Queries**

---

## [Editor Report · Acceptance letter]

12 Aug 2021

PGENETICS-D-21-00373R1 

The genetic consequences of dog breed formation - accumulation of deleterious genetic variation and fixation of mutations associated with myxomatous mitral valve disease in cavalier King Charles spaniels 

Dear Dr Axelsson, 

We are pleased to inform you that your manuscript entitled "The genetic consequences of dog breed formation - accumulation of deleterious genetic variation and fixation of mutations associated with myxomatous mitral valve disease in cavalier King Charles spaniels" has been formally accepted for publication in PLOS Genetics! Your manuscript is now with our production department and you will be notified of the publication date in due course.

With kind regards,

Andrea Szabo

PLOS Genetics

On behalf of:
